# Information theoretic evidence for layer- and frequency-specific changes in cortical information processing under anesthesia

Edoardo Pinzuti[1,2]*, Patricia Wollstadt[2], Oliver Tüscher[1,3,4], Michael Wibral[5]

**1** Leibniz Institute for Resilience Research (LIR), Mainz, Germany, **2** MEG Unit, Brain Imaging Center, Goethe University, Frankfurt/Main, Germany, **3** Department of Psychiatry and Psychotherapy, Johannes Gutenberg University of Mainz, Mainz, Germany, **4** Institute of Molecular Biology (IMB), Mainz, Germany, **5** Campus Institute for Dynamics of Biological Networks, Georg August University, Göttingen, Germany

* Edoardo.Pinzuti@lir-mainz.de

**Data Availability Statement:** The demo code and toolbox is available at: https://github.com/pwollstadt/IDTxl/tree/feature_spectral_ais. Raw LFP data are available from the Dryad Digital

## Abstract

Nature relies on highly distributed computation for the processing of information in nervous systems across the entire animal kingdom. Such distributed computation can be more easily understood if decomposed into the three elementary components of information processing, i.e. storage, transfer and modification, and rigorous information theoretic measures for these components exist. However, the distributed computation is often also linked to neural dynamics exhibiting distinct rhythms. Thus, it would be beneficial to associate the above components of information processing with distinct rhythmic processes where possible. Here we focus on the storage of information in neural dynamics and introduce a novel spectrally-resolved measure of active information storage (AIS). Drawing on intracortical recordings of neural activity in ferrets under anesthesia before and after loss of consciousness (LOC) we show that anesthesia- related modulation of AIS is highly specific to different frequency bands and that these frequency-specific effects differ across cortical layers and brain regions.

We found that in the high/low gamma band the effects of anesthesia result in AIS modulation only in the supergranular layers, while in the alpha/beta band the strongest decrease in AIS can be seen at infragranular layers. Finally, we show that the increase of spectral power at multiple frequencies, in particular at alpha and delta bands in frontal areas, that is often observed during LOC ('anteriorization') also impacts local information processing—but in a frequency specific way: Increases in isoflurane concentration induced a decrease in AIS in the alpha frequencies, while they increased AIS in the delta frequency range < 2Hz. Thus, the analysis of spectrally-resolved AIS provides valuable additional insights into changes in cortical information processing under anaesthesia.

## Author summary

While describing information processing in digital computers is somewhat straightforward and accessible (e.g. how much information is stored in a hard disk or which

Repository: https://datadryad.org/stash/dataset/doi:10.5061/dryad.z8w9ghxgp.

**Funding:** E.P. is currently employed by the Leibniz Institute for Resilience Research funded by the Leibniz Gemeinschaft. M. W. is employed at the Campus Institute for Dynamics of Biological Networks funded by the VolkswagenStiftung. O.T. is currently employed by the Leibniz Institute for Resilience Research and University Medical Center of the Johannes Gutenberg University Mainz and received funding by the German Research Foundation (DFG CRC 1193, subproject C04), the Ministry of Science of the state of Rhineland-Palatinate (LIR/DRZ program), and the European Union's Horizon 2020 research and innovation program (grant agreements No. 777084) in the context of this research. The funders had no role in study design, data collection and analysis, decision to publish, or preparation of the manuscript.

**Competing interests:** The authors have declared that no competing interests exist.

modification of information a CPU is executing), quantifying the widely distributed information processing in a biological neural system is much more challenging. In neural systems separating the components of distributed information processing—information transfer, storage and modification—helps with this task, but requires accurate mathematical definitions of these components of information processing. These definitions of distributed information processing quantities have become available only very recently. Of the three component processes mentioned above information storage, in particular, has been used with great success to analyze information processing in swarms, and to evolve, and optimize artificial information processing systems. The analysis of information storage has also already proven to be useful for the analysis of biological neural systems. Since in such systems, information processing seems to be often carried out by rhythmic neural activity with different frequencies, a measure of the frequency-specific components of the active information storage is needed. Here we introduce such a measure and study how isoflurane anesthesia affects the local information processing in the ferret prefrontal and primary visual areas around loss of consciousness. We found that the modulation of active information storage by isoflurane is specific to frequency, layers and area, and that the analysis of frequency-specific active information storage provides insights not captured by more traditional descriptions of neural activity.

## Introduction

Biological systems must process information about their environment and their internal states in order to survive. Many biological systems have evolved specialized areas where such information processing is particularly evident. Prime examples are the central nervous systems of many animals and the human brain in particular. Taking inspiration from such systems, humans have developed biologically-inspired, artificial information processing systems, such as artificial neural networks, to solve a variety of tasks. Artificial neural networks and their biological sources of inspiration share an important property—they perform highly distributed information processing in which fundamental information-processing operations such as storing, transferring and modifying information are both, highly distributed and co-located at almost all computational elements. The computational elements making up biological and artificial neural networks, for example, are neurons, where each neuron's activity can simultaneously serve the storage, transfer and modification of information. This lack of specialization and high degree of distribution separates such information processing systems from classical digital architectures (like a household PC) where the fundamental information processing operations are much more spatially separated and carried out by dedicated subsystems. While the highly distributed information processing certainly adds to the performance of artificial and biological neural networks on certain tasks, it also poses a formidable challenge to understand how such a system functions.

A powerful approach to describe and understand computation in systems such as biological or artificial neural networks is information theory, which introduces measures of information transfer, storage and modification [1–5]. The proposed measures are well-suited to investigate the function of artificial information processing systems, and have successfully been applied to biological neural systems [6–9]. However, in its original form, the framework neglects a central aspect of information processing in biological neural networks, namely the frequently displayed highly rhythmic activity when performing a computation. To understand those systems better and to build a bridge between information processing and their biophysical dynamics, it

would therefore be beneficial to link the components of information processing to specific neural rhythms.

We have recently presented such a link for the case of information transfer in [10], and have provided results that challenged some long-held *ad hoc* beliefs about the relationship of brain rhythms and information transfer. In the present work, we extend this approach to information storage. In particular we focus on the *active storage*, where the information storage is actively in use for a computation in the dynamic of the neural activity (for differences with passive storage, e.g synaptic gain changes, see [11]). A measure of this kind of storage is the *active information storage* (AIS) [3, 7], which quantifies the amount of information in the present samples of a process ("currently active") that is predictable from its past value. The AIS measure is closely linked to the transfer entropy (TE) [1]: the TE quantifies information transferred from a source process to the current value of a target process, in the context of the target process' own past. Hence, AIS and TE together reveal the sources of information which contribute to prediction of the target process' next outcome (either, information actively stored in the processes' own past, or additional information being transferred from another process) [7].

The importance of understanding how neurons and neural systems store information when studying neural information processing has been outlined already in [11] and later by the work of [3, 7, 12]. AIS as a measure of information storage has been successfully applied in magnetoencephalographic (MEG) recordings to test, for example, predictive coding theory [13] or to provide better understanding of the information processing in people affected with the Autism spectrum disorder (ASD) [6, 14]. In local field potential (LFP), [8] found an increased AIS measure as a function of anesthesia (isoflurane) concentrations in two ferrets recordings, at prefrontal (PFC) and visual cortical (V1) sites. Anesthetic agents such as isoflurane are known to affect the frequency spectrum throughout the cortex [15] and at laminar level [15, 16].

In [15] it was shown that the effect of isoflurane on neural oscillatory activity is not only frequency-specific but also related to the computational property of the area, being different between different areas of the cortex (PFC or V1) or between different layers (deep laminar or infragranular layers, granular layers, and superficial or supragranular ones). Similarly, [16, 17] reported highly specific effects of isoflurane on laminar frequency data.

Even though the effect of anesthesia on brain rhythms is known, due to a lack of a suitable method, all attempts to link the AIS with the rhythmic activity in different frequency bands were only indirect and through correlation analysis [6, 8, 14]. Hence, it seems beneficial to have also a spectrally-resolved AIS to directly investigate effects, for example, of isoflurane agents on brain rhythms and thus on neural information processing. We here present such a method, which is able to quantify AIS in a spectrally-resolved fashion.

We apply this method to laminar recordings from two areas of the ferret cortex (PFC and V1) under different levels of anesthesia, to investigate how different frequency bands contribute to information storage under anesthesia. We hypothesised that due to the different computational properties of the layers [18] (either deep or superficial), the frequency-resolved AIS would show a heterogeneity of anesthesia-related AIS changes across frequencies and recording sites. In more detail, the computational and oscillatory differences of AIS we expect are associated mainly to two distinct pathways that are responsible for communication between cortical areas and intracolumnar communication within area [17], i.e. feedfoward and feedback pathways. Feedfoward pathways are thought to carry sensory information from superficial layers to superficial and middle layers of higher cortical areas while feedback connections are thought to carry contextual information and predictions from deep layers to other deep or superficial layers of lower order areas [17–19]. Our choice of investigating the behaviour of

AIS in V1 and PFC is motivated by these areas being hierarchically well separated, with V1 at the bottom of the visual cortical hierarchy, and PFC being a hierarchically high association area. Investigating spectral AIS in ferrets is motivated by the fact that ferrets, as an intermediate model species, have similarities with primates, i.e. a highly developed visual system (V1) and cortical association areas such as PFC [15].

## Materials and method

### Ethics statement

Experimental procedures for ferrets cortical layers recordings were approved by the University of North Carolina-Chapel Hill Institutional Animal Care and Use Committee (UNC-CH IACUC) and exceed guidelines set forth by the National Institutes of Health and U.S. Department of Agriculture.

In this section, we first clarify the purpose and application of the proposed method. Second, we introduce the information theoretic preliminaries together with the AIS measure, and the corresponding notation. Central to our method is the creation of frequency-specific surrogate data, for which we summarize the technical background. Here, we outline only the crucial properties of the Maximal Overlap Discrete Wavelet Transform (MODWT), while a more detailed description can be found in [20, 21]. Finally, we present the core algorithm to identify frequency-specific AIS. Ferrets data employed in the AIS analysis can be obtained from the Dryad database [22].

### Background

**Problem statement and analysis setting.**   The aim of the proposed method is to determine whether there is statistically significant active information storage generated by one or more frequencies. Our method can be implemented after a significant AIS has been determined in the time domain, e.g. as computed by the AIS algorithm in [23], in order to provide a perspective on this novel spectrally-resolved AIS.

**Technical background: Active information storage (AIS).**   We assume that a stochastic process $\mathcal{Y}$ recorded from a system (e.g cortical or layers sites), can be treated as a realizations $y_t$ of random variables $Y_t$ that form a random process $\mathcal{Y} = \{Y_1..., Y_t, ..., Y_N\}$, describing the system dynamics. Then, *AIS* is defined as the (differential) mutual information between the future of a signal and its immediate past state [3, 7, 24]:

$$AIS(Y_t) = I(Y_t; \mathbf{Y}_{<t}), \tag{1}$$

where $Y$ is a random process with present value $Y_t$, and past state $\mathbf{Y}_{<t} = (Y_{t-\delta_1}, Y_{t-\delta_2} \ldots, Y_{t-\delta_k})$, with $\delta_i = i\Delta_t$, where $\Delta_t$ is the sampling interval of the process observation, and $\delta_1 \leq \delta_i \leq \delta_k$. $\mathbf{Y}_{<t}$ is a vector of random variables chosen from the process $\mathcal{Y}$ from the past of the current time point $t$. The collection, or vector, $\mathbf{Y}_{<t}$ captures the underlying dynamic of the process $\mathcal{Y}$ and can be seen as a state space reconstruction, for details see [3, 25]. We here employed a recently proposed non-uniform embedding algorithm from the IDTxl toolbox [23] to properly construct the nonuniform embedding of $Y$ time-series [26, 27]. This algorithm also yields approximations for parameters like $\delta$ and $k$. Thus, the AIS estimates how much information can be *predicted* by the next measurements of the process by examining its paste state [3]. In processes that either produce little information (low entropy) or that are highly unpredictable, the AIS is low, whereas processes that are predictable but visit many different states with equal probabilities [7], exhibit high AIS [7, 9].

**Technical background: Maximum overlap discrete wavelet transform.**   Our method is based on the creation of suitable surrogate data for use in a statistical test. Many methods exist for surrogate data creation, each with its own limitations and advantages (see [28] for a review). Among these, wavelet-based methods allow to create the needed frequency-specific surrogate data through randomization of the wavelet coefficients [29]. In particular, wavelet-based surrogates that preserve the local mean and the variance of the data were introduced by [30]. Similarly to [31], we employ the Maximal Overlap Discrete Wavelet Transform (MODWT), to transform the data in the wavelet domain. The MODWT is well defined for time-series of any sample size and produces wavelet coefficients and spectra unaffected by the transformation. [31].

The MODWT of a time-series $X = (X_0, \ldots, X_{N-1})$ of $J_0$ levels, where $J_0$ is a positive integer, consists of $J_0 + 1$ vectors: $J_0$ vectors of wavelet coefficients $\widetilde{\mathbf{W}}_1, \ldots, \widetilde{\mathbf{W}}_{J_0}$ and an additional vector $\widetilde{\mathbf{V}}_{J_0}$ of scaling coefficients, all with dimension $N$ (our exposition of the MODWT closely follows that of [20], pages 159–205). The coefficients of $\widetilde{\mathbf{W}}_j$ and $\widetilde{\mathbf{V}}_{J_0}$ are obtained by filtering $X$, namely:

$$\widetilde{W}_{j,t} = \sum_{l=0}^{L_j-1} \widetilde{h}_{j,l} X_{t-l \mod N}, \tag{2}$$

$$\widetilde{V}_{j,t} = \sum_{l=0}^{L_j-1} \widetilde{g}_{j,l} X_{t-l \mod N}, \tag{3}$$

where $\{\widetilde{h}_{j,l}\}$ and $\{\widetilde{g}_{j,l}\}$ are the $j$th level MODWT wavelet and scaling filters, with $l = 1, \ldots, L$ being the length on the filter and $L_j = (2^j - 1)(L - 1) + 1$. We can write the above in matrix notation as:

$$\widetilde{\mathbf{W}}_j = \underline{\widetilde{\mathcal{W}}_j} X \tag{4}$$

$$\widetilde{\mathbf{V}}_{J_0} = \underline{\widetilde{\mathcal{V}}_{J_0}} X \tag{5}$$

where each row of the $N \times N$ matrix of $\underline{\widetilde{\mathcal{W}}_j}$ has values denoted by $\{\widetilde{h}_{j,l}^{\circ}\}$, while $\widetilde{\mathcal{V}}_j$ has values denoted by $\{\widetilde{g}_{j,l}^{\circ}\}$, where $\{\widetilde{h}_{j,l}^{\circ}\}$ and $\{\widetilde{g}_{j,l}^{\circ}\}$ are the periodization of $\{\widetilde{h}_{j,l}\}$ and $\{\widetilde{g}_{j,l}\}$ to circular filter of length $N$ [20]. Thus, the MODWT treats $X$ as if it were periodic, such periodic extension is known as "circular boundary condition" [20]. Finally, the time series $X$ can be retrieved from its MODWT by [20]:

$$X = \sum_{j=1}^{J_0} \widetilde{\mathcal{W}}_j^T \widetilde{\mathbf{W}}_j + \widetilde{\mathcal{V}}_{J_0}^T \widetilde{\mathbf{V}}_{J_0} \tag{6}$$

While, the coefficients $\widetilde{\mathbf{V}}_{J_0}$ represent the unresolved scale [20, 31], and capture the long-term dynamics of $X$, the coefficients $\widetilde{\mathbf{W}}_j$ are associated with changes of the underlying dynamics, at a certain scale, over time. If $N = 2^J$ and we set $J_0 = J$, then a full transform is performed and the scale $\widetilde{\mathbf{V}}_{J_0}$ retains only the average constant of the data with all other information represented in the wavelet coefficients [31, 32]. Since in many applications a full transform is not necessary (e.g. the dynamic of a physical system is meaningful over a certain frequency range

only), $J_0$ can be set to any integer $J \leq \lfloor (\log_2(N)) \rfloor$ so that the transform at any scale is shorter than the total length of the time series [33]. The selection of $J_0$ determines the number of scales of resolution with the MODWT coefficients at a certain scale $j$ related to the nominal frequency band $|f| \in (1/2^{j+1}, 1/2^j)$ [20]. Moreover, given $\widetilde{\mathbf{W}}_j$ and $\widetilde{\mathbf{V}}_j$, it is possible to reconstruct the time-series $X$ through the inverse MODWT (IMODWT). If the coefficients are not modified, the IMODWT returns the original time-series $X$ [20]. As shown in [10] the MODWT is a suitable and efficient method to create surrogate data as required by the current algorithm.

## Algorithm

To obtain a frequency-resolved AIS measure, our algorithm's main idea is to create surrogate data, in which we destroy the AIS-relevant signal properties, i.e., the temporal order, in specific frequency bands. We then compare AIS estimates from the original data with estimates from the surrogate data, and establish via non-parametric statistical testing whether destroying specific frequency components led to a drop in AIS. This approach has has been successfully demonstrated in [10] to estimate frequency-specific TE and replaces approaches that use filtering or other preprocessing steps to estimate frequency-resolved measures, as these come with well known problems [10, 34]. As in [10], we here employed an invertible wavelet transform (maximum overlap discrete wavelet transform, MODWT) and a frequency- or scale-specific scrambling of the wavelet coefficients in time for surrogate data creation, keeping the original time-series always intact. With this method, we are also able to protect against false positive caused by a bias introduced by the wavelet filtering; this is because such a bias will only arise on the surrogate data, yielding a more conservative analysis. In other words, if the frequency-specific AIS measure should increase due to the scrambling of the wavelet coefficients, this will not result in a significant drop when statistically compared to the original AIS, and will thus not be mistaken for an effect.

**Implementation.**   Below, we will detail the algorithm for the measurement of frequency-specific AIS. As introduced above, we obtain this measure by creating surrogate data in which the temporal ordering of the signal has been destroyed for specific spectral components, by first transforming into the frequency domain, then scrambling wavelet coefficients and last transforming back to the time domain to obtain surrogate data. Overall this algorithm relies on five steps:

1. Perform a wavelet transform of the source time series through the MODWT to obtain a time-frequency representation of **Y** in $J_0$ scales.

2. At the $j$th scale of the MODWT transform shuffle the wavelet coefficients to destroy information carried by the scale (frequency band)

3. Apply the inverse wavelet transform, IMODWT, to get back the time representation of the time series

4. Compute the $AIS'_{freq}$ of the process **Y**.

   a. Repeat step 2 to 4 for a sufficiently high number of permutations to build a surrogate data distribution.

   b. Repeat step 1 to 4 for all $J_0$ scales.

5. Test whether the original $AIS$ is above the $1 - \frac{z}{J_0}$ quantile of the surrogate-based distribution of $AIS'_{freq}$ values at each scale, i.e. perform a significance test with respect to the surrogate-derived distribution.

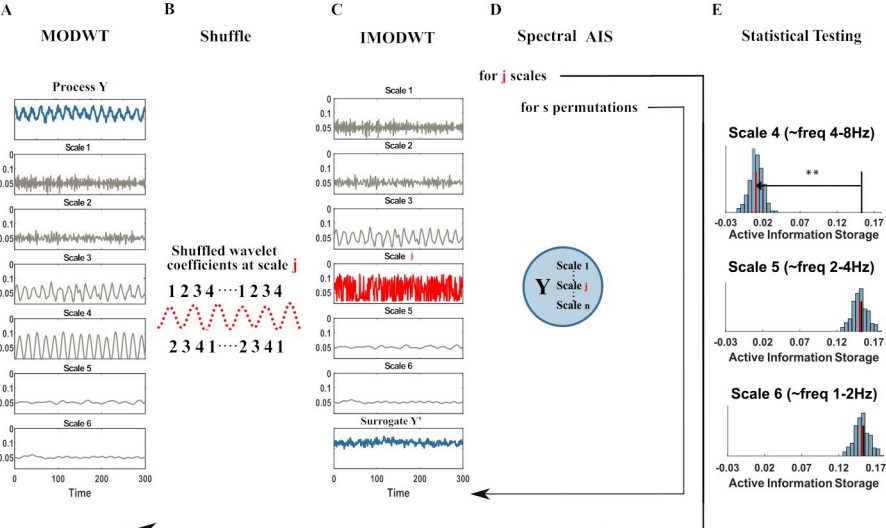

**Fig 1. Spectral AIS algorithm pipeline.** (A) The neural signal (blue) is converted to a time-frequency representation (grey) using the invertible maximum overlap discrete wavelet transform (MODWT). (B) At a frequency (wavelet scale) of interest in the source the wavelet coefficients are shuffled in time, destroying its internal dynamic. (C) The signal is recreated by the inverse MODWT. (D) The AIS for the original and many shuffled signals is computed. (E) A statistical tests determines whether the shuffling reduced the active information storage, indicating that the information storage was indeed encoded at the specific frequency. Each panel here shows the distribution of $AIS'_{freq}$ values (vertical bars) obtained from surrogate data where the wavelet coefficients of the scale of interest were shuffled, the median of this distribution (red line), and the original AIS (black line). The analysis and the testing is repeated for all scales of interest (here 4,5,6).

The operations implemented in the five steps are illustrated in Fig 1 and described in detail hereafter.

**Step 1**: The time-series is transformed once into $J_0$ scales through the MODWT (Fig 1A). As introduced in section *Maximum Overlap Discrete Wavelet Transform* this transform gives a set of coefficients $\widetilde{\mathbf{W}}_{1,...,J_0}$ and an additional set of approximation coefficients $\widetilde{\mathbf{V}}_{J_0}$. The latter is saved in this first step and utilized only in step 3, without any modification. Only the $\widetilde{\mathbf{W}}$ coefficients at the $j^{th}$ scale under analysis are subjected to step 2. The current implementation uses a Least Asymmetric Wavelet (LA) as mother wavelet of length 8 or 16, since both lengths showed to be robust against spectral leakage and do not relevantly suffer from boundary-coefficient limitations. [20, 30, 35].
The creation of surrogate data for subsequent statistical testing comprises of the following steps 2 and 3.

**Step 2**: The frequency-specific active information storage of the process is destroyed by shuffling the $\widetilde{\mathbf{W}}$ wavelet coefficients one scale at a time. The $j^{th}$ scale under analysis is shuffled by randomly permuting the coefficients $\widetilde{\mathbf{W}}_j$, whereas all the other scales transformed by the MODWT stay intact (Fig 1B $j^{th}$ scale in red). We implement two alternative methods for the creation of surrogate data: a Block permutation of the wavelet coefficients [29] and the Iterative Amplitude Adjustment Fourier Transform (IAAFT) [29, 31]. Since there is no canonical method of surrogate data creation and in many cases the employment of one method over another depends on the specific analysis carried out by the user.

**Step 3**: The unchanged set of coefficients, $\widetilde{\mathbf{W}}_{1,\dots,J_0\backslash j}$, the unchanged $\widetilde{\mathbf{V}}_{J_0}$'s, and the permuted coefficients at scale $j$ ($\widetilde{\mathbf{W}}_j$) are submitted to the IMODWT, to reconstruct the surrogate process signal, $\mathbf{Y}'$, in the time-domain (Fig 1C). This step is identical for both of the implemented surrogate-data creation methods: Block permutation of the wavelet coefficients and IAAFT. The reconstructed process $\mathbf{Y}'$ (*process surrogate*) differs from the process $\mathbf{Y}$ only on the shuffled $j^{th}$ scale. In this way, we destroy the process information storage only if is carried by the $j^{th}$ scale, otherwise the information storage stays the same.

**Step 4**: With $\mathbf{Y}'$ we compute again the *AIS*. We illustrated this step in Fig 1D. Let $\mathbf{Y}_{<t}$ be the set of past variables of the *process* previously found in the analysis, with $\mathbf{Y}'_s$ being the $s$-th *process surrogate* under analysis at scale $j$; then, the $AIS'_j$ for the surrogate data is:

$$AIS'_j = I(Y_{t,s}; \mathbf{Y}'_{<t,s}), \tag{7}$$

The algorithm is repeated from step 2 to step 4 for *s permutations*, with $s = 1, \dots, S$, to create a distribution of surrogate $AIS'_{j,s}$ values; $S$ is set according to the desired critical level for statistical significance (including Bonferroni correction for the number of scales, see below). Subsequently, all the $J_0$ scales transformed by the MODWT in step 1 are subjected to step 2, step 3 and step 4, such that $J_0$ separate distributions of $AIS'_{j,s}$-values, one for each scale, are obtained.

**Step 5**: As a final step, the *AIS* is tested for statistical significance against the $J_0$ different distributions of $AIS'$ surrogate values. If the $\mathbf{Y}^j$ (where $j$ is one of the scales transformed by the MODWT) contributes to the generation of the active information storage in the process $\mathbf{Y}$, a significant drop of the $AIS'_j$ surrogates will be observed. This step is applied for all $J_0$ scales under analysis and a Bonferroni correction is applied such that each individual scale is tested at the significance level $\alpha/J_0$.

Additionally, each scale analyzed is plotted, see Fig 1E, and we restrict ourselves to interpret only the scale that shows maximal distance (or well separated local maxima) from the original *AIS*, $\max_j(AIS - \widetilde{AIS'_j})$, where $\widetilde{AIS'_j}$ denotes the median of the surrogates distribution at scale $j$. We consider the maximal distance in addition to the statistical significance test because frequency transform is never perfect (e.g. due to leakage, noise and overlapping wavelet bands). Indeed, validation of the algorithm on synthetic data shows that the maximum distance reliably reflects the ground truth, whereas the statistical significance test can suffer from leakage effects on adjacent scales. Obviously, this limits the detectability of frequency-specific *AIS* and may be overly conservative. Thus, in scenarios, where AIS from multiple frequency bands is strongly expected *a priori*, or where the length of the data allows for vanishing leakage effects, the above restriction may be lifted.

## Results

In the following section we test the capability of the proposed algorithm to recover frequency specific AIS. To this end, we employed three simulations, where the ground truth is known. These simulations are limited to three example cases only, because the core idea and implementation strictly followed the spectral TE algorithm [10] (see above). For this more complex case of source-target interactions we have already demonstrated in depth that the MODWT construction of frequency specific surrogates in combination with a suitable statistical test reliably delivers a frequency resolved information theoretic measure [10].

In addition to the proof-of-principle on simulated systems, we applied the spectrally resolved AIS on Local Field Potential (LFP) data from ferrets under different levels of isoflurane and at recording sites in different cortical, and at sites in the prefrontal cortex (PFC) and in primary visual cortex (V1). For each combination of cortical area and layer we assessed if the AIS and the frequency resolved AIS were modulated as a function of different isoflurane concentrations using Bayesian linear regression.

All the analysis of AIS and spectrally resolved AIS below were performed with a block permutation of the wavelet coefficients (for construction of surrogates) and LA(8) as mother wavelet, similarly to [10].

### Example I: Null case, no information storage

At first, we simulated the case of no AIS in a process, to evaluate the behavior of our algorithm when none of the frequency scales generates information storage. We employed a white noise process, which by definition should not contain any information storage. However, we point out that, the spectral AIS is a post-analysis step, which can be applied only after significant AIS is found in the time-domain. We repeated the null-case simulation scenario 500 times to estimate the number of false positive results. In the time-domain the number of false positive results was below the alpha level ($alpha$ = 0.05), while the spectral AIS analysis revealed no false positive result (see Fig 2, panel A), indicating that the strategy to exploit wavelet filtering for surrogates data creation is robust. Sometimes, surrogate data show AIS values that are consistently larger than in the original data in certain bands (see Fig 2, panel A, scale 1 and 2). This is an illustration of the unwanted distortions created by spectral processing of the data. If we processed the original data, e.g. via filtering, this would be of great concern, as it would lead to false positives. In our algorithm, however, it only makes our method slightly more conservative. This is also supported by our test of the empirical false positive rate (see Fig 2, panel A).

### Example II: Specificity test

Secondly, we assessed the specificity of the spectral AIS analysis to demonstrate that the method reveal no significant AIS to all bands except the ones of interest. Thus, we simulated a signal as the sum of two sinusoids: one with oscillations at $50 Hz$ and one with oscillations at $12 Hz$, for 10 s with a sampling rate of $120 Hz$ and 50 trials. This simulation should show a significant AIS in the first scale (i.e. frequency band 30–60$Hz$) and at the third scale (i.e. frequency band 7.5–15$Hz$) but no spectral AIS drops at the intermediate scale(i.e. frequency band 15–30$Hz$). As before, we repeated the analysis 500 times to estimate the number of false positive for the specificity test. As can be seen in Fig 2, panel B, only scale 1 and 3 (the scales of interest, shaded gray box) showed a significant drop, while all other bands did not revealed a significant result, indicating that the test is specific. The number of false positive results in the time domain was below the alpha level ($alpha$ = 0.05).

### Example III: Fractionally integrated AR process

Third, we simulated the combination of a fractionally integrated process with an autoregressive process (AR). The resulting process belongs to the class of fractionally integrated autoregressive moving average process (ARFIMA) [36, 37]. The simulated process exhibits autonomous oscillation at $f_1$ = 50 Hz, long-term memory and positive information storage [37]. The AR amplitude was set to $p$ = 0.98 and the differencing parameter was set to $d$ = 0.3, generating 10 s at sampling rate of 120 Hz and 100 trials. The coefficients of the AR process

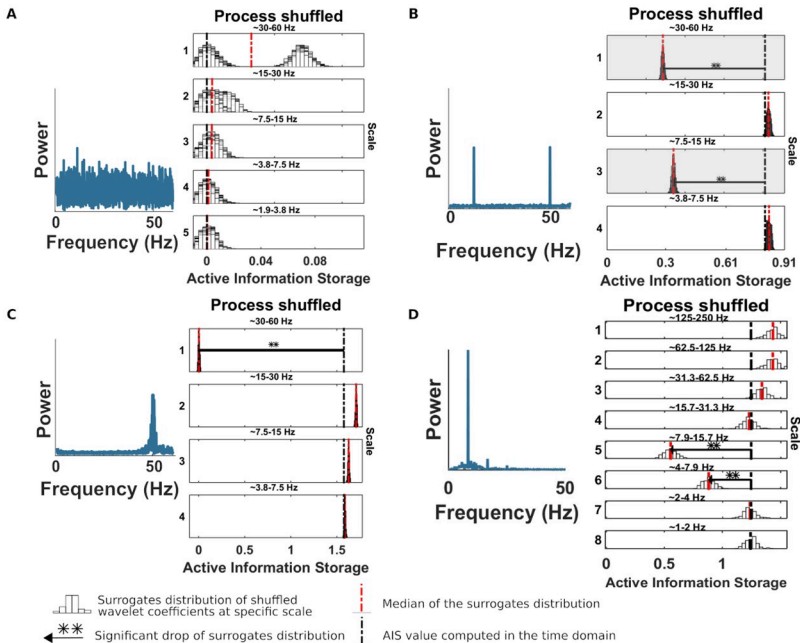

**Fig 2. Spectrally-resolved AIS for three exemplary simulations.** Each panel, shows the *AIS′* distribution obtained from the surrogate data with shuffled coefficients at the scale indicated to the left, or, equivalently, the frequency band indicated at the top of each panel. White bars represent histograms of surrogate data, i.e. relative frequencies in (a.u.), the red dashed line is the median of the surrogate *AIS′* distribution, the black dashed line is the original AIS value. The horizontal black line indicates the distance between the original AIS and the median of the surrogate distribution (**, $p < 0.005$; *, $p < 0.05$) Panel A, spectral AIS for the null-case example (500 simulations). Left, the power spectra of white noise process. Right, spectrally resolved AIS at different scales (frequency bands). No significant drop of the shuffled wavelet coefficients can be found since the process had no internal dynamic. For this simulation red dashed line is the median of the the mean of the surrogates across the 500 simulations. Panel B, specificity test of the spectral AIS. Left, the power spectra of the signal. Right, spectrally resolved AIS at different scales. No significant drops can be seen in all bands except the one of interest (shaded gray box). Panel C spectral AIS for example 3, linear. Left, power spectra of the ARFIMA process with spectral peak at 50 Hz. Right, spectrally resolved AIS at different scales (frequency bands). The AIS showed, correctly, a significant drop at scale 1 (30–60Hz). Panel D, spectral AIS for example 4, nonlinear. Left, power spectra of the realizations of a selected variable of the Rossler system, with a spectral peak at around 8 Hz. Right, spectrally resolved AIS at different scales (frequency bands). The AIS showed, correctly, the largest drop at scale 5 (8–16 Hz) and scale 4 (4–8 Hz).

were generated with the following equation:

$$A_1 = 2p\,cos(2\pi f_1) \tag{8}$$

$$A_2 = -p^2 \tag{9}$$

while the fractionally integrated AR process is generated with the following equation:

$$A(L)(1 - L)^d X_n = U_n \tag{10}$$

where the coefficients A are generated with Eqs 8 and 9, U is a Gausssian white noise with zero mean and unit variance and $(1 - L)^d$ is the fractional differencing operator computed by the fast fractional difference algorithms [38]. First, we analysed the process in the time domain to establish the presence of significant AIS. Then, we applied the spectrally-resolved AIS algorithm to obtain the frequency information of the system. Correctly, the scale 1 (frequency band $30--60 \sim$ Hz), containing the spectral peak at $50 \sim$ Hz, shows a significant drop of AIS in the surrogate data set indicating spectral AIS at that scale (see Fig 2, panel C).

### Example IV: Chaotic dynamical system oscillator

In this fourth simulation, we evaluated the spectral AIS with a process generated by a non-linear dynamical system that exhibits self-sustained periodic oscillations, similar to [10, 39]. The system was simulated with the following equation:

$$\frac{dx_1}{dt} = -w_1 y_1 - z_1 + \epsilon x_2(t - \tau) \qquad \frac{dx_2}{dt} = -w_2 y_2 - z_2$$

$$\frac{dy_1}{dt} = w_1 x_1 + 0.15 y_1 \qquad\qquad \frac{dy_2}{dt} = w_2 x_2 + 0.15 y_2 \qquad (11)$$

$$\frac{dz_1}{dt} = 0.2 + z_1(x_1 - 10) \qquad \frac{dz_2}{dt} = 0.2 + z_2(x_2 - 10)$$

where $w_1$ and $w_2$ are the parameters governing the natural frequencies of the oscillators, which were set to 0.8 and 0.9, and $\epsilon = 0.07$ is the coupling strength and $\tau$ is the time delay, which was set to 2 time steps. Additionally, Gaussian white noise was added to the generated time-series. The analysis was performed on the assumption that only variables $x_1(t)$ could be observed. As can be seen in Fig 2, panel D, the process $x_1(t)$ oscillated around 8 Hz. The sampling rate was 500 Hz, and 25 trials of length 4 seconds were generated (100000 samples).

As before, first we established the presence of significant AIS in the time domain. Then, we obtained the spectral AIS as in the examples before. The results indicate that the largest drop was at scale 5, with also a significant drop at scale 6, which was expected as the frequency of the process spanned both scales (see Fig 2, panel D).

### Spectral AIS under anesthesia at different cortical layers via Bayesian regression

We applied our spectral AIS method to electrophysiological recordings of LFP data. Data were recorded in V1 and PFC in two different female ferrets, at supragranular layers, the granular layer and infragranular layers, under different concentrations (0.5%, 0.75%, 1%) of isoflurane and under awake conditions (0%).

These laminar LFP data have been analysed previously in terms of frequency spectrum modulations at different isoflurane concentration in [15]. Here, we provide a spectrally-resolved assessment of the AIS in these signals, we hypothesise that AIS is modulated by isoflurane concentration in a layer- and brain-region-specific way. All methodological and recording details can be found below and in [15].

### Electrophysiological recordings and pre-processing

Recordings were made in adolescent female ferrets (15–20 weeks old) that had not reached sexual maturity to exclude possible estrous-dependent changes in physiology [15]. Each recording session was conducted in a dark room during resting state, lasting no more than two hours during which animals's head were fixed. Anesthetize recordings utilized varying concentration of isoflurane anesthesia with xylazine (iso: 0.5%, 0.75%, 1%). All three concentrations of isoflurane anesthesia corresponded to a lack of behavioral responses. Twenty minutes elapsed after anesthetic concentration were changed. Two linear 16-channel silicon probes (100-$\mu$m contact site spacing along the z-axis; Neuronexus, Ann Arbor, MI) were used in cases of dual craniotomies. A silver chloride wire tucked between the skull and soft tissue and held in place with 4% agar in saline was used as the reference for both linear probes. Each probe was slowly advanced into cortex with a micromanipulator (Narishige, Tokyo, Japan); correct depth was determined by small deflections of the LFP at superficial electrode recording sites

and larger deflections of the LFP at deeper electrode recording sites. Unfiltered signals were first amplified with MPA8I head stages with gain 10 (Multichannel Systems, Reutlingen, Germany) and then further amplified with gain 500 (model 3500; A-M Systems, Carlsborg, WA), digitized at 20 kHz (Power 1401; CambridgeElectronic Design, Cambridge, UK), and digitally stored using Spike2 software (Cambridge Electronic Design). For analysis, data were low pass filtered (300 Hz cutoff) and down-sampled to 1000 Hz. Data were visually inspected and segments of data were removed, if they were contaminated by artifacts. At the conclusion of the study, all animals were humanely killed with an overdose of pentobarbital sodium and immediately perfused with 4% formaldehyde in 0.1 M phosphatebuffered saline for subsequent histological verification of recording locations. Additional information of animals surgery can be found in [15]

**AIS and spectral AIS estimation.** First, we estimated the AIS from LFP recordings. We implemented a similar approach as in [8], using the IDTxl toolbox [23] to determine the presence of significant AIS value at the layer level. To make any claim about isoflurane concentration effects we had to use identical embeddings (see Section: *Technical background: Active information storage (AIS)*) for the estimation of the AIS or spectral AIS measures at different isoflurane levels in order to equilibrate the estimation bias across these levels. To this end, we applied the following four analysis steps:

1.  Run the AIS algorithm for each trial (length 8 seconds) and isoflurane level.

2.  Take the union of all embeddings across trials and isoflurane levels (i.e. the union of all past state variables identified).

3.  Compute the AIS measure using the union embedding.

4.  Apply the union embedding for the spectral AIS algorithm and quantify the frequency (scale/frequency band) contribution as:

$$AIS_{freq}^{\Delta} = AIS - \widetilde{A}IS' \tag{12}$$

where, AIS is the original measure computed in the time domain and $\widetilde{A}IS'$ is the median of the AIS distribution estimated on surrogate data with coefficients shuffled at a specific frequency scale. Thus, the $AIS_{freq}^{\Delta}$ reflects the contribution of the particular frequency band under analysis to the AIS. Only positive values correspond to a significant contribution to the formation of AIS in the process under investigation.

**Bayesian linear regression layers: Model specification.** For the analysis of LFP laminar data, we employed a Bayesian linear regression model. The dependent variables were the *AIS* and $AIS_{freq}$. For the *i*th trial, we can define the likelihood of the AIS measure as:

$$y_i \sim \mathcal{N}(\alpha + \beta x_{i,iso}, \sigma^2) \tag{13}$$

where $\alpha$ is the intercept and encodes the mean AIS, the parameter $\beta$ is the slope which captures the isoflurane experimental effect, whereas the term $x_i^{iso}$ encodes the isoflurane levels (0%, 0.5%, 0.75%, 1%), and $\sigma^2$ is the residual variance.

We choose a Normal distribution as a *prior* for the parameters $\alpha$ and $\beta$ and Halfnormal distribution for the $\sigma$ parameter, all values for the parameters of the *prior* distributions can be found in S1 Table. We built one model (we refer to this model as "simple model") for each layer at PFC site and V1 site, separately (6 models in total). Since the plotted data showed a possible quadratic effect as a function of different isoflurane concentrations, we additionally

built the same six models with an added quadratic term (we refer to this model as "quadratic model"), so that the likelihood terms become:

$$y_i \sim \mathcal{N}(\alpha + \beta x_{i,iso} + \beta_{sq} x_{i,iso}^2, \sigma^2) \tag{14}$$

with a Normal distribution as a prior for $\beta_{sq}$ (see S1 Table). Finally, for each layer we evaluated the model that predicted the data better (simple model vs quadratic model) using a leave-one-out cross-validation (LOO-CV) score as outlined next.

**Bayesian regression setup and model comparison.**    We estimated the model regression coefficients using Bayesian inference with Markov Chain Monte Carlo (MCMC) sampling, using the python package pymc3 [40] with NUTS (NO-U-Turn Sampling), using multiple independent Markov Chains. We implemented four chains with 3000 burn-in (tuning) steps using NUTS. Then, each chain performed 10000 steps, those steps were used to approximate the posterior distribution. To check the validity of the sampling, we verified that the R-hat statistic was below 1.05.

To evaluate different models with different numbers of parameters, we implemented cross-validation, which has been advocated for Bayesian model comparison, e.g. in [41]. In particular we adopted the LOO-CV implemented in PyMC3. Lower LOO-CV scores imply better models. We report the full modeling and model comparison results in supplementary tables: S2–S9 Tables, and only include the results of the winning models in the main text.

**Hierarchical bayesian regression.**    We point out that an alternative modeling approach to asses the anesthesia effects on the AIS measure would have been adopting a Hierarchical Bayesian Regression [42]. In a hierarchical model, parameters can be viewed as a sample from a population of parameters; for our case this implies to set a hyperprior from which we sample the $\beta$ parameters for the two cortical areas (PFC and V1) and three layers (infragranular, granular and supragranular). This modeling approach would be optimal in case of the prior assumption of a certain amount of similarity of the AIS behaviour between these cortical structures, or in other words, an overall effect common to the different cortical areas and layers. The Bayesian framework allows to include such prior knowledge on the model formulation. However, previous work on spectral power [15, 16], as mentioned above, showed that anesthesia modulates cortical areas and layers differently. Based on this prior knowledge we decided to model each single layer in each cortical area (PFC and V1) separately, resulting in six separate models.

**Cortical layer and brain-region specific modulation of total AIS by isoflurane.**    We start by reporting the result of the Bayesian regression analysis for the *AIS* dependent variable, in the time domain, and subsequently the result of the frequency resolved AIS (*AIS*$_{freq}$).

First, we evaluated the AIS measure for different isoflurane levels in the time domain, and performed the Bayesian regression analysis.

In V1, the models with a squared beta coefficient described the data better than the models without it, as indicated by the LOO-CV-based Bayesian model comparison [41] (lowest LOO score in supplementary S2 Table). In contrast, in PFC, the two types of models were almost indistinguishable; yet the *quadratic model* performed slightly better as well (see supplementary S2 Table).

In PFC, in the infragranular layer, we found a consistent increase of the AIS as a function of isoflurane concentration (yellow line) with a posterior mean of *beta iso* = 1.7, [1.28, 2.13] and *beta iso squared* = 0.9, [0.52, 1.27] (Fig 3, panel A) and also in the granular layer of PFC *beta iso*: = 0.48, [−0.075, 1.05] and *beta iso squared* = 1.43, [0.92, 1.91] (Fig 3, panel B), while for the supragranular layer of the PFC the effect of isoflurane on AIS was minimal, with a posterior

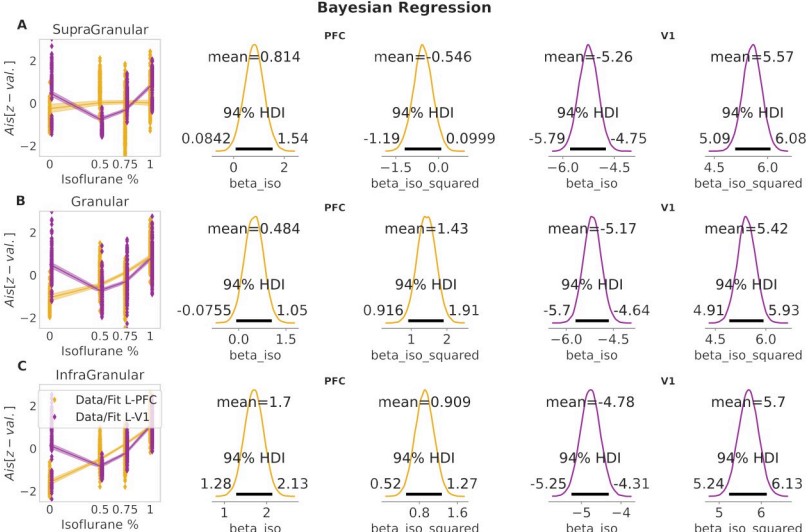

**Fig 3. Bayesian regression results of spectral AIS in the time-domain.** Each column show the Bayesian regression fit for V1 (purple) and PFC (yellow) at supragranular (rigth), granular (middle) and infragranular (left). Shaded area in the regression fit around the estimated mean (solid line) represents 94% HDI.

mean of *beta iso* = 0.814, [0.08, 1.54] and *beta iso squared* = −0.54, [−1.19, 0.099] (see Fig 3, panel C).

In V1, all layers showed a similar behavior (see Fig 3, panels A–C), with a decrease of AIS values for intermediate isoflurane concentrations (0.5% and 0.75%) and a subsequent increase for the highest isoflurane level (1%). Posterior means for *beta iso* were = −4.78, [−5.24, −4.3], −5.17, [−5.69, −4.64] and −5.25, [−5.77, −4.73], for infragranular, granular and supragranular layers, respectively (see Fig 3, panel A–C). Similarly, posterior means for the beta coefficient of the squared isoflurane concentration were close to each other with *beta iso squared* = 5.7, [5.24, 6.13], 5.42, [4.91, 5.93], 5.57, [5.09, 6.08], for infragranular, granular and supragranular layers, respectively (see Fig 3, panel A–C).

In summary, deeper layers in PFC (infragranular and granular) showed stronger modulation under increasing isoflurane levels compared to supragranular layers, such a clear difference did not appear between layers in V1. This result is in line with [8], where a more pronounced increase of AIS (at increasing isoflurane concentrations) was found at PFC compared to V1.

**Cortical layer and brain-region specific modulation of frequency-specific AIS by isoflurane.** Next, we evaluated the $AIS_{freq}$ for different isoflurane levels in the frequency domain in multiple frequency bands. We start presenting the results for $62.5Hz–125Hz$, i.e. the high gamma frequency band.

In this band, in PFC, the *quadratic model* was substantially better than the *simple model* in the infragranular and granular layers; in the supragranular layer the *quadratic model*, despite still fitting the data better than the *simple model*, only had a marginally better LOO score (see supplementary S3 Table).

In V1 at the infragranular layer, LOO scores for the models with or without the *beta iso squared* coefficient were almost identical, whereas for granular and supragranular layers the *quadratic model* represented a better description of the data by the model (see S3 Table).

In the high gamma band ($62.5Hz–125Hz$) we observed a modulation of the $AIS_{freq}^{\Delta}$ mostly in the supragranular layer of V1 (see Fig 4 last panel, top and Fig 5, panel G), with an increase for

## Bayesian Regression of spectral AIS

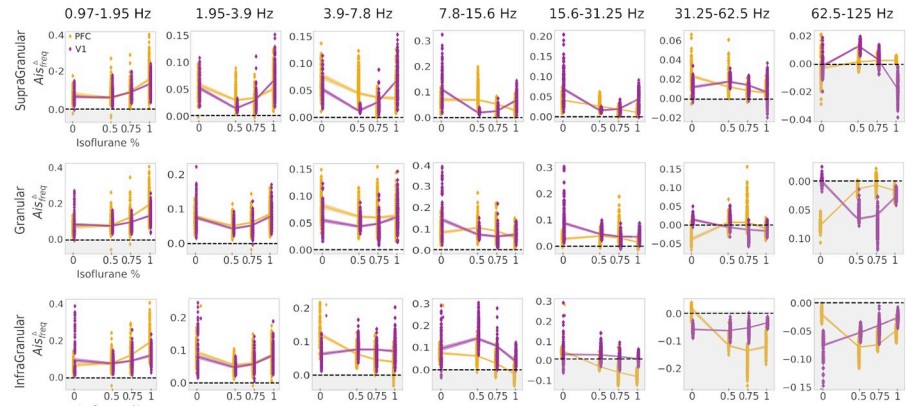

**Fig 4. Bayesian regression results of spectral AIS in the frequency range 0.95–125Hz.** Each column show the Bayesian regression fit at specific frequency range for V1 (purple) and PFC (yellow) at supragranular (top row), granular (middle row) and infragranular (bottom row). Shaded area in the regression fit around the estimated mean (solid line) represents 94% HDI. Shaded gray background for $AIS_{freq}^{\Delta}$ values that are below zero (i.e. no frequency specific drop).

intermediate levels of isoflurane (0.5%) *beta iso* = −0.92, [−0.96, −0.89] and *beta iso squared* = −0.86, [−0.89, −0.83] and a subsequent decrease for higher isoflurane concentrations, whereas in PFC such a modulation was absent in supragranular layer. Modulation was also absent at both, V1 and PFC, in granular and infragranular layers (see Fig 4, last column, middle and bottom panels).

In the frequency range 31*Hz*–62*Hz* (i.e. gamma band), the *quadratic model* was better for all layers at both brain regions (V1 and PFC). Nevertheless, at the granular layer of V1 and at the supragranular layer of PFC the LOO-CV difference with the *simple model* was minimal (see supplementary S4 Table).

Similarly to the high gamma band, the supragranular layer was the layer most pronouncedly modulated by the different isoflurane concentrations. In the PFC the $AIS_{freq}^{\Delta}$ decreased as a function of isoflurane *beta iso* = −0.48, [−0.54, −0.41] and *beta iso squared* = −0.09, [−0.04, −0.14], while in V1 it increased for isoflurane at 0.5% followed by a decrease *beta iso* = −0.43, [−0.51, −0.35] and *beta iso squared* = −0.42, [−0.5, −0.36] (see Fig 4, sixth column, top panel and Fig 5, panel F). At granular layer of V1, the positive $AIS_{freq}^{\Delta}$ values for isoflurane at 0% decrease to negative (see Fig 4, sixth column, middle panel, shaded gray background) for intermediate and high level of isoflurane concentrations (from 0.5% to 1%). Finally, no relevant modulation could be seen in the infragranular layers of both PFC and V1 brain areas (see Fig 4, sixth column, bottom panel). Taken together, the results for gamma and high gamma band, showed an isoflurane effect on $AIS_{freq}^{\Delta}$ mainly in the superficial layer (supragranular) and minimally in the granular layer, in agreement with association of gamma band to superficial layers [43], see Section: *Modulation of spectral information storage according to distinct functional roles across cortical layers by anesthesia*, for further details.

In the frequency range 15*Hz*–31*Hz* (i.e. beta band), the *simple model* had a lower LOO-CV score for the supragranular layer of PFC. In all other cases, the *quadratic model* had lower LOO-CV score (see S5 Table).

In this frequency band, in PFC, the supragranular and infragranular layers decreased as isoflurane levels increased. While in the supragranular layer $AIS_{freq}^{\Delta}$ value were still positive at

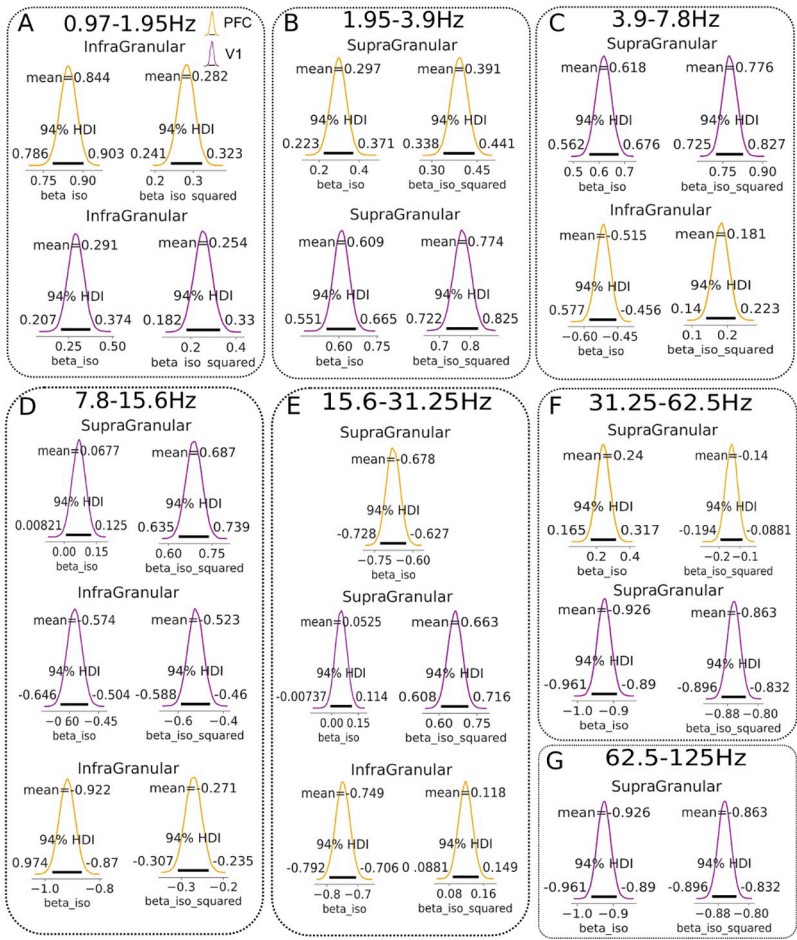

**Fig 5. Selection of Bayesian posterior distributions.** From panel A to panel G, Bayesian posterior distributions with highlighted mean of the posterior distribution for *beta iso* and *beta iso squared* coefficients and 94% HDI. Each panel contained a selection of relevant Bayesian analysis results at different cortical layers for PFC (purple) and V1 (yellow).

isoflurane 1%, *beta iso* = −0.67, [−0.72, −0.62], in the infragranular layer the $AIS^{\Delta}_{freq}$ values became negative at higher isoflurane levels (0.75% and 1%), *beta iso* = −0.74, [−0.79, −0.70] and *beta iso squared* = 0.11, [0.08, 0.16], revealing that the spectral surrogate drop was abolished (see Fig 4, fifth column, bottom panel, shaded gray background and Fig 5, panel E).

In V1 the supragranular layer had a different modulation, compared to PFC, with a significant decrease at isoflurane 0.5% and a subsequent increase for isoflurane 0.75% and 1%, *beta iso* = 0.05, [−0.007, 0.11] and *beta iso squared* = 0.66, [0.60, 0.71] (see Fig 4, fifth column, top panel and Fig 5, panel E). In granular layer after a decrease for concentration 0.5%, $AIS^{\Delta}_{freq}$ remained in a similar range of values for high isoflurane concentrations (see Fig 4, fifth column, middle panel).

In the frequency range 8*Hz*–15*Hz* (alpha/beta band), the *quadratic model* had a lower LOO-CV for all layers at both PFC and V1 regions (see S6 Table). Nevertheless, for the supragranular layer at PFC and granular layer of V1 the models had overlapping standard errors of the estimates (SE).

In this frequency range, in V1, we observed a similar behavior of supragranular and granular layers to the previous frequency range, posteriors for *beta iso* = −0.06, [−0.008, −0.125] and

*beta iso squared* = 0.687, [0.63, 0.73] of supragranular layers were very close to the ones in the frequency range 15*Hz*–31*Hz* (compare values of Fig 5, between panel D and E). In contrast, the infragranular layer of V1 had a opposite modulation to that found in the next higher frequency range with an increase for intermediate isoflurane level (0.5%) and a later decrease at higher isoflurane concentrations, *beta iso* = −0.574, [−0.64, −0.50] and *beta iso squared* = 0.52, [0.588, 0.46] (see Fig 5, panel D).

In PFC the $AIS_{freq}^{\Delta}$ was mainly modulated from 0.5% to 1% isoflurane levels and this modulation was strongest in the infragranular layer (compare Fig 4, fourth column; top, middle and bottom panels). Despite a common shift of alpha power from posterior to anterior cortex during loss of consciousness (LOC) [44], none of the layers at PFC showed an $AIS_{freq}^{\Delta}$ component increase. We discuss the absence of the alpha anteriorization effect (for the AIS) in Section: *Modulation of spectral information storage according to distinct functional roles across cortical layers by anesthesia*.

In the frequency range 4*Hz*–8*Hz* (theta band), the *quadratic model* was substantially better only in the supragranular layer of V1 (see S7 Table); in all other cases the difference with the *simple model* was minimal, yet the *quadratic model* had a nominally lower LOO-CV score.

In PFC, we observed a modulation of the $AIS_{freq}^{\Delta}$ by isoflurane, in all three layers (see Fig 4, third column, top, middle and bottom panels. The strongest decrease was for infragranular and supragranular layers with values: *beta iso* = −0.515, [0.57, −0.45] and *beta iso squared* = 0.18, [0.14, 0.22] for infragranular layer (see Fig 5, panel C).

In V1 the supragranular layer had a strong decrease for isoflurane 0.5% but this was followed by an increase for higher isoflurane levels *beta iso* = 0.618, [0.56, 0.67] and *beta iso squared* = 0.776, [0.725, 0.827] (see Fig 5, panel C).

In the frequency range 1.95*Hz*–4*Hz* (delta band), the models with a squared beta coefficient described the data better than the models without it in all layers at PFC and V1, as indicated by LOO cross-validation- based Bayesian model comparison (lowest LOO-CV score, see S8 Table).

At this low frequency band the modulation by isoflurane became more homogeneous across layers and brain regions. Indeed, we found that the infragranular and granular layer of PFC and V1 were all similarly modulated (see Fig 4, second column, top, middle and bottom panels). Only in supragranular layers the increase for high isoflurane values was stronger in V1 compared to PFC, with *beta iso* = 0.609, [0.55, −0.66] and *beta iso squared* = 0.77, [0.72, 0.82] for V1 and *beta iso* = 0.297, [0.22, 0.37] and *beta iso squared* = 0.391, [0.33, 0.44] (see Fig 5, panel B).

Finally, we estimated the frequency range 0.9*Hz*–1.9*Hz* (low delta band). The Bayesian model comparison revealed that the *quadratic model* had a lower LOO score in all the layers, however in the granular layer of V1 standard error of estimates overlapped, indicating that both models fitted the data similarly (see S9 Table).

As in the previous frequency range, deep and superficial layers at both brain regions were characterized by a similar isoflurane modulation, due to a possible global effect of slow oscillation throughout the cortex under LOC (see Section: *Modulation of spectral information storage according to distinct functional roles across cortical layers by anesthesia*). However, in this frequency range, infragranular and granular layers at PFC had a stronger increase at high isoflurane concentrations (0.75% and 1%) than in V1, with the highest difference in the infragranular layer *beta iso* = 0.84, [0.78, 0.903] and *beta iso squared* = 0.284, [0.24, 0.32] for PFC and *beta iso* = 0.291, [0.20, 0.37] and *beta iso squared* = 0.25, [0.18, 0.33], while supragranular layer showed a similar modulation at both brain regions (see Fig 4, first column, and Fig 5, panel A for values of the bayesian mean of the posterior). All posterior distributions can be found in supplementary figures: S1–S7 Figs.

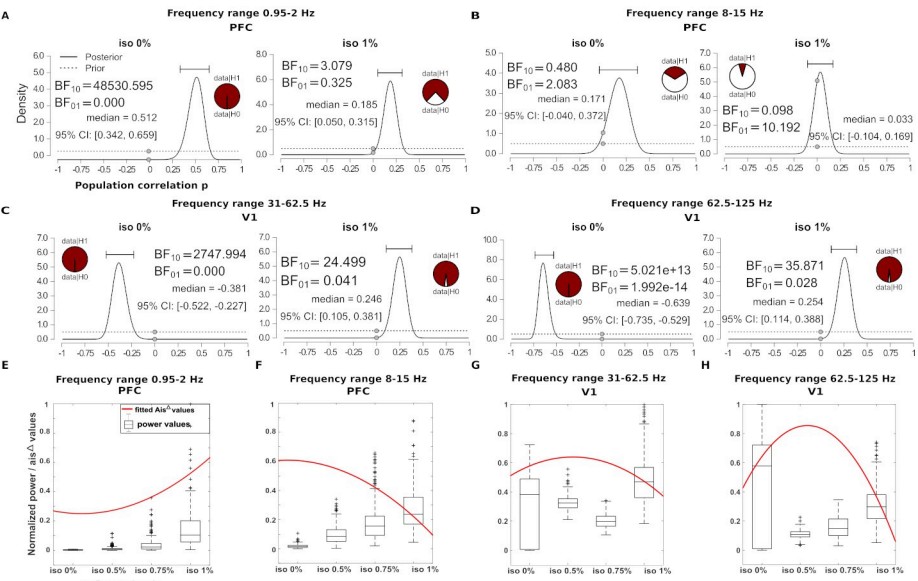

**Fig 6. Bayesian correlation of spectral AIS and spectral power.** (A to D) Bayesian correlation of spectral AIS with spectral power in different frequency ranges at isoflurane 0% and 1% in PFC and V1. Each panel shows the posterior distribution (black line), prior distribution (dashed line), median and 95% of the posterior estimates of the regression coefficients and the Bayes factor BF10 in favor of the alternative hypothesis (H1) relative to the null hypothesis (H0), where $BF10 = p(D|H1)/p(D|H0)$; BF01 is just the inverse of this, and is given for convenience only. (E to H) Modulation of AIS (red curve) and spectral power (black box-plot) with isoflurane levels, in different frequency ranges at PFC and V1. (Scatter plots with the correlation fit can be found in S8 Fig).

**Relation between the spectral AIS and the spectral power.** To further highlight the differential behaviour of spectral AIS compared to spectral power we performed a Bayesian correlation of spectral power in two frequency ranges with spectral AIS: $0.9Hz$–$1.9Hz$ (delta) and $7.9Hz$–$15Hz$ (alpha) at infragranular PFC and in the frequency ranges $31Hz$–$62.5Hz$ (gamma) and $62.5Hz$–$125Hz$ (high gamma) of supragranular V1. The Bayesian correlation revealed that the correlation between the two measures varied depending on the frequency band and isoflurane level. In the frequency range $0.9Hz$–$1.9Hz$ at isoflurane of 0% there was a strong evidence for a positive correlation $BF_{10} = 48530.5$ (see Fig 6, panel A, right); at isoflurane of 1% the correlation dropped to moderate evidence $BF_{10} = 3.079$ (see Fig 6, panel A,left). In the alpha band $7.9Hz$–$15Hz$ at isoflurane of 0% the evidence for a correlation was absent $BF_{10} = 0.4$ (see Fig 6, panel B, right), while at isoflurane of 1% there was moderate evidence for the null hypothesis $BF_{01} = 10.19$ (see Fig 6, panel B, left). Additionally, we showed that as the power increases as a function of isoflurane the $AIS_{freq}^{\Delta}$ had opposite behavior; the frequency specific surrogates drop increased at higher isoflurane percentage in the range $0.9Hz$–$1.9Hz$, while it decreased in the range $7.9Hz$–$15Hz$ (see Fig 6, panel E and F). In V1, gamma and high gamma frequencies showed a similar behaviour. At isoflurane of 0% there was strong evidence for a negative correlation (see Fig 6, panel C and D, right) which became a positive correlation at isoflurane of 1% (strong evidence, see Fig 6, panel C and D, left). Interestingly, the relation with the spectral AIS indicated an inverted u-shape, with increased predictability (higher AIS values) at intermediate isoflurane concentration and lower AIS values at isoflurane of 1% (see Fig 6, panel G and H). We discuss the increased gamma predictability in Section: *Modulation of spectral information storage according to distinct functional roles across cortical layers by anesthesia.*

To further characterize the difference between spectral power analysis and AIS measure, we performed an additional simulation. We generated three different signals with the following

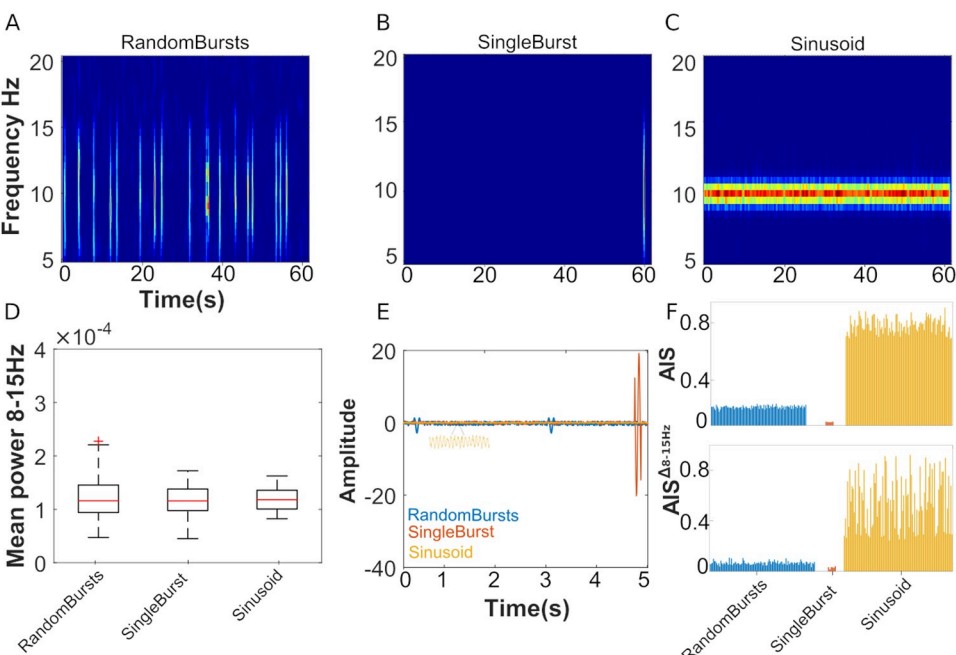

**Fig 7. Simulation of different oscillatory patterns and related AIS estimation.** Panel A, example of a signal with randomly spaced bursts around 10$Hz$ placed on Gaussian noise, panel B, example of a signal with a single burst placed randomly at the end of a Gaussian noise, panel C a sinusoid with Gaussian noise at 10$Hz$. Panel D, shows the mean power, for 100 simulations of the three scenarios, in the frequency range 8–15$Hz$. Panel E, an example of the three simulated time-series (last five seconds). Panel F, results of the AIS and spectral AIS analysis for the three tested simulated signals.

features: randomly spaced oscillatory bursts around 10$Hz$, with random phases, interspersed by Gaussian noise (see Fig 7, panel A, top row), a single oscillatory burst randomly placed at the end of a Gaussian noise time-series (see Fig 7, panel B, top row) and a sinusoid with additional Gaussian noise (see Fig 7, panel C, top row). We generated 100 examples of these signals (for the sinusoid the frequency component varied within 8–15$Hz$), and scaled the signals to obtain a similar mean power in the frequency range 8–15$Hz$ (see Fig 7, panel D, bottom row), which is the resolution of AIS spectral power analysis (7.5$Hz$–15$Hz$ for scale 3, at the sampling frequency of 120$Hz$). Because of the different dynamic of the three constructed signals, we show that for the random and single burst signals the AIS and the spectral AIS (for the frequency band 7.5$Hz$–15$Hz$) is lower compared to the sinusoid, despite having similar power (see Fig 7, panel E, bottom row). Indeed, the past of the signal for randomly spaced bursts contains minimal information to predict its future, similarly with the single burst (which show significant AIS only when burst at the end of the signal is covered by the past state of the AIS $Y_{<t}$, compared to the high predictability of a sinusoid.

This simulation helps to interpret the result of the different behavior of the power and spectral AIS in Fig 7 panel D, in favour of a more random bursting activity in the range 7.9$Hz$–15$Hz$ at high isoflurane concentrations, and higher power at the maxima of these bursts. Rhythmic activity can be transient in non-averaged data [45] and has been observed in different frequency bands: gamma, beta and alpha; and isoflurane has been associated with burst suppression in the alpha band [46]. This scenario would explain a higher total mean mean power and a reduction of the spectral AIS (see Fig 7 panel D). We further corroborate this hypothesis with an additional alpha-beta burst analysis [45]. We identified bursts in the frequency range 8–15$Hz$ and showed that the power maxima of these bursts increase at higher

isoflurane concentrations (see S10 Fig, panel C). Additionally, the comparison of the coefficient of variation ($CV^2$) of the inter-event intervals with the Fano Factor (that quantifies the trial-to-trial-variability of events number per trial), where the event is the alpha-beta burst, indicated that at isoflurane 0.75% and 1% the underlying process for the bursts generation might be a Poisson process (see S10 Fig, panel D), which would explain the observed low AIS. Detailed information of the burst analysis is reported in supplementary information: S1 Text

## Discussion

In this study we addressed two tightly related questions: 1. Can we in principle design an algorithm to detect frequency-specific active information storage? 2. Do we detect such frequency-specific active information storage in neural systems, and if so, does it provide valuable information towards a better understanding of neural information processing?.

### Estimating spectrally-specific AIS

To address the first question, we presented an algorithm to estimate which specific spectral components contribute to the overall active information storage (AIS) of a process. We demonstrated with different simulations that these spectral components can be reliably identified, in both linear and nonlinear systems and processes. The algorithm builds on the idea of creating spectrally specific realizations of the null model (surrogate data) that was presented in [10], for the case of spectrally-resolved TE. In the present study, we again used the MODWT decomposition and scrambling of the wavelet coefficients for the creation of spectrally-specific surrogates data to assess frequency contribution to the AIS measure.

The spectrally-resolved AIS can be seen as an attempt to determine frequency components related to information-theoretic properties of a process under investigation. Its estimation is less complex than the estimation of spectral TE which implies to decompose specific spectral components of sources and targets and to distinguish one-to-one from one-to-many or many-to-one interactions, whereas spectral AIS provides spectral resolution only of a single process. Thus, it is expected that the proposed algorithm performs similarly well as the previously published one for spectral TE. We, therefore, keep the discussion of the validity and performance short here.

### Which insights into neurophysiology can be obtained using spectral AIS?— The example of anesthesia effects

The second question becomes crucial in biological systems such as the brain where the role of rhythmic processing in neural system is still not fully understood but prominently discussed for neural communication mechanisms [47–49]. We note that for AIS as a foundational component of neural information processing, only a handful of studies exist to date [6–8, 13], and none has asked the question of its relation to neural rhythmic processing.

The analysis of LFP cortical layers at two brain sites (PFC and V1) in ferrets, showed that we can successfully detect frequency-specific AIS at different frequency bands. While in the time domain the total AIS showed an increase as a function of isoflurane level, the spectral perspective revealed a much more complex and richer picture, where AIS rose in certain bands but decreased in others with increasing isofluorane levels. Furthermore, in comparing modulations of spectral AIS and spectral power we showed that AIS provides information on the computational dynamics of the neural process and its modulation by anesthesia, which spectral power analysis does not reveal.

In the remainder of this section, we will further discuss the results of the application of our novel, spectrally-resolved AIS measure to the LFP under anesthesia, highlight the additional

details that the spectrally-resolved AIS revealed compared to time-domain-only analysis of AIS and the additional information that the spectral AIS provides compared to spectral power analysis also in terms of anesthesia effects. We place the LFP spectral AIS results in the context of our current understanding of cellular and circuit mechanisms of anesthesia in order to shed some light on how cortical computation changes under anesthesia.

Last, we will then indicate limitations and caveats of the method, discuss its relation to previous approaches and possible future applications.

**Modulation of spectral information storage according to distinct functional roles across cortical layers by anesthesia.** Due to the multiple dimensions along which our AIS values change, i.e. cortical areas, cortical layers, anesthesia levels, and frequencies, we have to focus our discussion here on the most prominent findings. For these most prominent findings, we will try to highlight possible functional consequences based on current theories of cortical function, and to suggest neurophysiological mechanisms underlying the observed changes in spectral AIS where this seems possible. An important reference point to keep in mind for the the discussion below is the fact that the animals have already lost behavioral responsiveness at an isoflurane level of 0.5% [15]. Thus, changes in spectral power and spectral AIS beyond this point are unlikely to be central to anesthesia-induced loss of consciousness.

**Effects in frequencies below the theta band.** One previous observation linking anesthesia and active information storage [8] was the overall increase of time-domain AIS with increasing depth of anesthesia, which was also observed here across cortical layers and areas (but most prominently in PFC, see Fig 3). The analysis of spectral-AIS showed that these increases in AIS are driven by frequencies below the theta band. Combined with the increase in spectral power under anesthesia that is frequently described for these very low frequency bands [15], our findings indicate the presence of *highly regular*, high amplitude low frequency activity. Here, the well known high amplitude of low frequency activity suggests large scale spatial synchronization whereas the high regularity found via spectral-AIS analyses indicates stereotypically repeating activity; together, these findings point to a greatly reduced richness of cortical processing—possibly not enough to sustain consciousness. This is compatible with an earlier work of Bharioke and colleagues who found greatly reduced entropy in layer 5 pyramidal cells after the onset of high-amplitude, low frequency oscillations [50]. At the biophysical level the emergence of high-amplitude low-frequency oscillations has been linked to a decoupling in the cortico-thalamico-cortical loop (as discussed in [50–52]).

**Spectral AIS in the beta and alpha band.** In contrast to the above low-frequency activity, spectral power changes in the beta and alpha bands have been frequently linked to the awake state and the performance of cognitive tasks [53]. Neurophysiologically, alpha and beta-band activity have been linked to spatio-temporally structured inhibition (e.g. [54, 55]), the maintenance of a cortical status quo [56], and feedback signalling of internal predictions [13, 57] the loss of beta-band activity under administration of certain drugs, such as ketamine, has been linked to phenomena of distorted perception [58] that precede the loss of consciousness. The observed loss of spectral AIS in these bands under isoflurane in our study—sometimes despite increases in spectral power (Fig 6 and Fig 2 in [15])– points to a loss of temporal structure (also see supplementary S10 Fig). This temporal structure, however, may be necessary for some or all of the above cortical functions. Given that the alpha and beta bands have been linked to the maintenance of internal predictions, their loss of temporal structure may lead to a degraded internal model of the world, i.e. a loss of an organized representation of the world around us, and also our internal bodily state. It is conceivable that such a loss of an internal representation of the world is one component of the phenomenon of loss of consciousness. Biophysically, changes in alpha and beta-band activity under anesthesia may be linked to a disruption of the cortico-cortical and cortico-thalamic

loop [59] via anesthesia effects on layer 5 pyramidal neurons. In essence, anesthesia decouples the distal apical dendrites, that receive feedback inputs, from the somatic compartment in infragranular-layer pyramidal neurons resulting in a widespread decoupling of both, cortico-cortical and cortico-thalamo-cortical feedback throughout cortex [59]. This may make the sustained representations of internal models impossible, in line with what was said above based on spectral AIS results.

**Gamma band activity.**   Traditionally, gamma band activity has been linked to numerous cognitive functions ([18, 53, 60], but is also sometimes just seen as an indicator of overall cortical activity [61]. In predictive coding theories, gamma band activity is sometimes linked to the anatomical feedforward signalling of prediction errors [13, 18]. When taking into account the predominant origin of high-frequency gamma band oscillation in supragranular cortical layers [16] and the anatomical feedforward signalling from projection neurons in the supragranular layers [62] it seems plausible to assume that gamma oscillations are related more to signalling towards the core of the cortical processing hierarchy. If so, the content represented in activity in the gamma band should be more variable and less predictable. As a consequence gamma-band activity should be linked to a lower spectral AIS than alpha and beta band activity. This is indeed what we observed: gamma-band AIS was significant only in superficial layers, and generally lower than spectral AIS in the alpha and beta band—as expected. Therefore, we tentatively interpret the observed changes in gamma band AIS in relation to feedforward signalling and prediction errors. The observed increases in gamma-band AIS in V1, i.e. lower in the hierarchy, at an isofluorane level of 0.5% coincide with a loss of spectral power in both gamma bands (see Fig 6). This seems compatible with the notion that signals in these bands to contain a priori unpredictable error-related activity. A loss of this activity would thus increase predictability in these bands (spiking activity decreased at this concentration in [15]). As error-related signalling towards the core of the cortical hierarchy, however, may be also a prerequisite for a meaningful adaptation of internal models, the loss of unpredictable gamma band activity might also mean the loss of the ability to adapt internal models to the outside world.

**Link to neurophysiology.**   We are aware that the above links of anesthesia-related changes in spectral AIS to anesthesia-related loss of consciousness heavily depend on assumptions borrowed from predictive coding theories. Our interpretation is inspired for example by [52], who describe putative links between predictive coding theories and theories of conscious processing, and link these to the pivotal role of layer 5 pyramidal cells. In these cells, the apical dendrites in particular are prime mediators of anesthetic effects brought about by a loss of their inputs from thalamus [59]; isoflurane anesthetic upregulate $GABA_A$ receptors [63] and downregulates high-order thalamic nuclei [52, 59]. The loss of thalamic inputs to the apical dendrites renders these cells incapable of fusing bottom-up and top-down processing streams in cortex—in the context of predictive coding theories this is compatible with a loss of the ability to adapt an internal model to changing sensory inputs, and ultimately to maintain an internal model altogether. Our above interpretation of computational changes related to changes in spectral AIS points to a similar mechanism based on computational considerations.

We thus describe an interpretation of spectral AIS changes and their relation to anesthesia and loss of consciousness that is coherent with both, predictive coding theories and proposed biophysical mechanisms of anesthesia. Nevertheless, additional research needs to substantiate or refute predictive coding theories in general, and their description of anesthesia and loss of consciousness in particular.

**Spectrally-resolved AIS provides insights into neural processing and the effects of anesthesia that are not provided by an analysis of spectral power.**   Anaesthetic agents such as isoflurane, sevoflurane or propofol produce similar oscillatory changes, in particular,

predominating low frequencies (delta band) and increased power in the alpha frequency band at frontal sites (anteriorization effect) [44, 64, 65]. We show here that the spectral AIS provides additional information on the underlying neural information processing and the effects of isoflurane. For example, we describe effects in the alpha frequency band that are not found by an analysis of only spectral power: while at low frequencies (delta band) and no isoflurane (0%), the spectral AIS and the spectral power are strongly correlated, such a relation can not be seen in the alpha band (see Fig 6A and 6B, right panel). Additionally, while in the delta band the spectral AIS follows the increase of spectral power as a function of isoflurane concentration, an opposite behaviour can be seen in the alpha band (see Fig 6E and 6F). Interestingly, the alpha frequency shifts (increase of anterior alpha power and decrease of posterior alpha power) showed the same spectral AIS profile (compare Fig 6, panel F of PFC with S9 Fig, panel B of V1), thus the observed power shifts seem to be independent of the spectral AIS. In the frequency range $8-15 Hz$ we performed an alpha-beta burst analysis which resolved in a possible explanation for the decrease of the spectral AIS in this frequency range (see S10 Fig). Even though, we take this result with caution (this type of analysis could be affected by the burst identification method), deep general anesthesia has being associated with burst suppression in this oscillatory range [46, 66]. The burst suppression of the alpha rhythm is characterized by periods of high voltage activity (burst) and flatline segments, almost periodic, but with inter- and intra burst variations [67]. This phenomenon occurs only at deep level of anesthesia (see Fig 1 in [66]), and may arise as the interaction between neural dynamics and brain metabolism [66]. Interestingly, this effect at only high isoflurane doses (0.75% and 1%) seems to be present also in our data (see S10 Fig, panel B), while the underlying bursts generation at these isoflurane doses seems to be related to a Poisson process (Fano factor an $CV_2$ almost 1, see S10 Fig, panel D).

For other anesthetics such as propofol, recent work showed that alpha band effects depended on two types of thalamocortical circuits affected by the anaesthetic agents and were completely distinct from the propofol-induced slow oscillations [65], as well as computational model point to such distinction [68]. Thus, we speculate that also propofol-related changes in spectral AIS will be distinct for the delta- and the alpha-bands. Hence, decomposing the AIS measure in its spectral components can reveal aspects of the computational dynamics of neural processes that are not directly accessible by a spectral power analysis (see Fig 7, of how different oscillatory patterns give rise to AIS).

**Spectrally-resolved AIS adds additional insights into the effects of anesthesia on neural processing compared to AIS in the time domain.**   In a previous study analyzing LFPs from ferrets under anesthesia, the AIS (in the time domain) increased as a function of isoflurane concentrations in PFC [8]. Given the similar effect that we found in this work in the time domain (see Fig 3, panel C), the overall increase at high isoflurane levels in AIS seems to be linked to an increase in AIS in delta frequencies, whereas alpha and beta frequency bands are modulated by isoflurane differently, i.e. they decrease as a function of isoflurane levels. Alpha and beta bands have been linked to generation of internal models in the predictive coding framework [13], and have also been associated mostly with deep cortical layers, and thereby, cortical feedback pathways [16]. Thus, the absence of alpha and beta-band AIS may suggest— following the line of argument in [13]—that under anesthesia the maintenance of internal models and the generation of internal predictions is strongly impaired. This in turn may be an important component of the phenomenon of loosing consciousness, indeed shutting the coupling in the pyramidal neurons and integration of inputs from superficial to deep layers of the cortex would lead to a drastic breakdown in the cause-effect repertoire and consciousness would fade [52, 69, 70]

## Relation to previous approaches

A method that allows to compute AIS at different temporal scales has been introduced by [71]. It exploits the state space formalism to obtain a multiscale representation of a linear fractionally integrated autoregressive process (ARFI) [71]. The time-series undergo to lowpass filtering and downsampling to obtain a multiscale representation, so that the AIS can be computed as a function of the cutoff frequency. This parametric formulation, employing the state space formalism is restricted to the description of linear Gaussian processes. However, it is a significant improvement over previous attempts to quantify system complexity in terms of a linear multiscale entropy [72], with the simultaneous description of short and long memory properties which are fundamental aspects of systems dynamic [37]. Another potential method comes from the equality of Granger causality and information transfer or information theoretic measure for Gaussian variables in the time and frequency domain [73, 74]. Also, recent work showed that for linear Gaussian processes the information modification (one of the component of information processing) may be formulated, analytically, in the frequency domain [75, 76] as the synergy component of a partial information decomposition following the idea of [77], and using the PID measure $I_{\mathrm{MMI}}$ of [78]. This frequency decomposition approach for linear Gaussian processes can be also adopted for the AIS. When the assumptions of linear Gaussian processes are valid, then the methods in [71] or [75, 76], will be more data-efficient and come with lower computational burden. Furthermore, the frequency estimation of the AIS for Gaussian variables will then allow for a more precise identification of the relevant frequency components (the work in [75, 76], evaluates the all frequency spectrum while the current method can only identify frequency bands). If the assumptions of linear Gaussian processes are not met however, our approach seems to be the only viable alternative at the moment.

## On the possibility of cross-spectral information storage

Due to the sensitivity of information-theoretic measures to non-linear phenomena it is conceivable to find information storage in cases where the frequency of the process underlying the storage changes over time, i.e. where the stored information wanders between frequencies as the process unfolds. If, for example, the information is moving forth and back between dynamics at certain low frequencies and certain high frequencies, this should be detectable, using the approach from [10]. Thus, as an extension to the algorithm presented here, it is possible to destroy the information in a specific frequency also in the future of a process instead of the past, similar to the individual frequency-specific destruction of information in source and target processes in the estimation of TE [10], and to thus find which frequency is the past source of information, and which frequency is the current target of the information transported from the past into the future. This way, effects of nonlinear dynamics in a process maybe made visible. Although this is not used in the current manuscript, it is implemented in $IDT^{xl}$ [23], for future investigations.

## Caveats and limitations

The estimation of information-theoretic quantities, such as the AIS, from finite data is highly non-trivial (e.g. [79] and references therein). In many cases the necessary number of physical realizations of a process is not available. Two possible strategies can be implemented then: pooling data over time to obtain a sufficient amount of realizations (this requires stationarity) or pooling data over an ensemble of temporal copies. This latter approach approach exploits the cyclostationarity across these temporal replications of the process. Last, for discrete-valued data, Bayesian approaches exist for optimization embedding parameters and AIS estimation [80]; these approaches are available in our Toolbox [23].

### Future directions

Future studies should focus on combining spectrally resolved transfer entropy [10] and active information storage to provide a more exhaustive characterization on the computational behaviour of the analysed system in the spectral domain. Employed together, these tools offer a promising framework to test specific hypothesis on brain functioning such as predictive coding theory [9] or encoding and maintenance of information in working memory [81]. For example, frequency-resolved measures of information transfer and active information storage can test specific hypothesis on LFP-frequency signatures of error signals [18, 43] or coding of prior information [13]. Similarly, maintenance of relevant information for later reactivation, in working-memory and prefrontal cortex has been associated with specific frequency signature [81]. Also here our spectrally resolved algorithms can, thus, provide additional insights on the relation between brain rhythms and information-processing.

Information modification is the third important component next to information transfer and storage. Thus it is an obvious question to ask, if we can also equip an analysis of information modification with spectral resolution. To see why this exceedingly difficult it is good to recall that information modification has been linked to an information theoretic quantity called synergy, resulting from a partial information decomposition ([77]). Thus, ideally one would like to isolate this synergistic component of the joint mutual information between two (or, potentially, more) source processes and a target process, then apply surrogate data creation by spectral processing as demonstrated for mTE [10] and AIS (here), and show how the synergy is distributed across the various combinations of spectral components. At present we deem the construction of such a spectrally resolved measure of information modification difficult, and especially difficult to interpret, as the presence of multiple bands in the sources and in the target already leads to a problem related to partial information decomposition [10]. Thus, interpreting the results of such an analysis would mean to keep track of multiple information components of to nested partial information decompositions. Combining this insight with the current state of the field of partial information decomposition, where multiple, different concepts of redundancy, synergy, and unique information are still in the processes of being defined based on their applicability to various task settings and questions (operational interpretations) makes the endeavour of defining a spectral measure of information decomposition seem premature at the moment.

## Conclusion

In this study we have presented an algorithm that provides a spectral representation of the computational dynamics of neural processes in terms of the active information storage. Using this algorithm for the analysis of changes in neural information processing under anesthesia, we showed that this analysis can add valuable additional insights that are not provided by the analysis of changes in spectral power.

Our method is fully available and integrated in the open source package $IDT^{xl}$: https://github.com/pwollstadt/IDTxl/tree/feature_spectral_ais, along with a demo script.

## Supporting information

**S1 Table. Priors of the Bayesian linear regression models.**
(PDF)

**S2 Table. Results of LOO-CV model comparison for AIS in the time domain.**
(PDF)

**S3 Table. Results of LOO-CV model comparison for AIS at 62.5Hz–125Hz.** (PDF)

**S4 Table. Results of LOO-CV model comparison for AIS at 31Hz–62Hz.** (PDF)

**S5 Table. Results of LOO-CV model comparison for AIS at 15.6Hz–31.2Hz.** (PDF)

**S6 Table. Results of LOO-CV model comparison for AIS at 7.8Hz–15.6Hz.** (PDF)

**S7 Table. Results of LOO-CV model comparison for AIS at 4Hz–7.8Hz.** (PDF)

**S8 Table. Results of LOO-CV model comparison for AIS at 1.95Hz–4Hz.** (PDF)

**S9 Table. Results of LOO-CV model comparison for AIS at 0.9Hz–1.9Hz.** (PDF)

**S1 Fig. Bayesian regression results of AIS at frequency 62.5–125Hz.** Panel A, left, Bayesian regression fits for supragranular layer at PFC (yellow) and V1 (purple). Panel B, left, Bayesian regression fits for granular layer at PFC (yellow) and V1 (purple). Panel C, left, Bayesian regression fits for infragranular layer at PFC (yellow) and V1 (purple). Middle columns, posterior mean for *beta iso* and *beta iso squared* coefficients at PFC site, for panel A, B and C. Right columns, posterior mean for *beta iso* and *beta iso squared* coefficients at V1 site. Shaded area in the regression fits represents 94% HDI. Shaded gray background for $AIS^{\Delta}_{freq}$ values that are below zero (i.e. no frequency specific drop). (TIFF)

**S2 Fig. Bayesian regression results of AIS at frequency 31.25–62.5Hz.** See S1 Fig for display conventions. (TIFF)

**S3 Fig. Bayesian regression results of AIS at frequency 15.6–31.25Hz.** See S1 Fig for display conventions. (TIFF)

**S4 Fig. Bayesian regression results of AIS at frequency 7.8–15.6Hz.** See S1 Fig for display conventions. (TIFF)

**S5 Fig. Bayesian regression results of AIS at frequency 3.9–7.8Hz.** See S1 Fig for display conventions. (TIFF)

**S6 Fig. Bayesian regression results of AIS at frequency 1.95–3.9Hz.** See S1 Fig for display conventions. (TIFF)

**S7 Fig. Bayesian regression results of AIS at frequency 0.97–1.95Hz.** See S1 Fig for display conventions. (TIFF)

**S8 Fig. Bayesian correlation of spectral AIS and spectral power.** Bayesian correlation of spectral AIS with spectral power in different frequency ranges.
(TIFF)

**S9 Fig. Spectral power and spectral AIS relation in V1.** Panel A, the modulation of AIS (red curve) and spectral power at isoflurane levels, in the delta band (black box-plot). Panel B, the modulation of AIS (red curve) and spectral power at isoflurane levels, in the alpha band (black box-plot).
(TIFF)

**S10 Fig. Bursts analysis in the alpha/beta band.** Panel A, Pearson's correlation coefficient between mean alpha/beta power and the percent of pixels in the spectrogram above cutoff in the non-averaged spectrogram. Various median cutoffs were calculated and plotted on a log scale. Black line correlation is the mean across isoflurane concentrations. Panel B, example of alpha/beta burst detection for each isoflurane concentration (single trial). White dot denotes the local maxima in the spectrogram, with maxima power above 7x median power cutoff. Panel C, Boxplot denotes the mean frequency power at the burst maxima, for different isoflurane concentrations. Panel D, Fano Factor and $CV^2$ for each isoflurane concentrations.
(TIFF)

**S1 Text. Alpha-beta bursts analysis pipeline.**
(PDF)

## Acknowledgments

We are grateful to K. Seller for the LFP ferrets recordings and F. Fröhlich to make the data available.

## Author Contributions

**Conceptualization:** Edoardo Pinzuti.

**Data curation:** Edoardo Pinzuti.

**Formal analysis:** Edoardo Pinzuti.

**Funding acquisition:** Oliver Tüscher.

**Investigation:** Edoardo Pinzuti.

**Methodology:** Edoardo Pinzuti, Patricia Wollstadt, Michael Wibral.

**Project administration:** Oliver Tüscher, Michael Wibral.

**Resources:** Michael Wibral.

**Software:** Edoardo Pinzuti, Patricia Wollstadt.

**Supervision:** Oliver Tüscher, Michael Wibral.

**Visualization:** Edoardo Pinzuti.

**Writing – original draft:** Edoardo Pinzuti, Michael Wibral.

**Writing – review & editing:** Edoardo Pinzuti, Patricia Wollstadt, Oliver Tüscher, Michael Wibral.

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
