## [Decision Letter · Decision Letter 0]

20 Sep 2022

Dear Mr Pinzuti,

Thank you very much for submitting your manuscript "Information theoretic evidence for layer- and frequency-specific changes in cortical information processing under anesthesia" for consideration at PLOS Computational Biology.

Sorry for the delay in this first decision.

As with all papers reviewed by the journal, your manuscript was reviewed by members of the editorial board and by several independent reviewers. In light of the reviews (below this email), we would like to invite the resubmission of a significantly-revised version that takes into account the reviewers' comments.

Please take particular care in situating the method and the results in the state of the art, simplifying the exposition and stating the novelty and the relevance.

We cannot make any decision about publication until we have seen the revised manuscript and your response to the reviewers' comments. Your revised manuscript is also likely to be sent to reviewers for further evaluation.

Sincerely,

Daniele Marinazzo

Section Editor

PLOS Computational Biology

Reviewer's Responses to Questions

**Comments to the Authors:**

Reviewer #1: The article by Pinzuti et al. proposes an approach to assess frequency-specific storage of information in neural systems, with application to cortical information processing under anesthesia. The method builds on previous work of the same group in the context of (i) estimating active information storage (AIS) [e.g., Ref. 7] and (ii) estimating frequency-specific information transfer using wavelet-transform surrogate data [Ref. 10]. Despite this incremental aspect, the proposed methodology is original as it fills a gap in the current literature in the field, and its application showing the effects of anesthesia on neural information processing convincingly shows the usefulness of the approach (e.g., when comparing frequency-specific AIS to time-domain AIS and to spectral power). In view of this, and considering that the neurophysiological findings are original and the paper is well written, I recommend acceptance of the manuscript after addressing the comments in the following.

Recent studies have shown that the two other components of information processing, i.e. information transfer and information modification, in the particular case of linear Gaussian processes, can be formulated analytically in the frequency domain and related to their time-domain counterparts in an intuitive way [e.g., Chicharro, D. (2011), Biological cybernetics, 105(5), 331-347; Faes, L., et al. (2021, Philosophical Transactions of the Royal Society A, 379(2212), 20200250; Antonacci, Y., et al. (2021), IEEE Access, 9, 149486-149505]. These works are relevant to the present paper, because similar formulations could be provided also for the information storage, and therefore it would be interesting to compare them, at the level of the discussion, with the approach proposed here and in Ref. [10] to assess frequency-specific information storage and information transfer.

Also in view of the previous comments, the Authors could mention in the subsection about “Future directions” the extension of their method to the evaluation of information modification, also highlighting the potential issues related to its implementation.

Another aspect that could be mentioned in the discussion is that the measure of information storage, computed in a model-free way as done in this work, reflects also the nonlinear interactions between rhythms manifested in a signal at different frequencies; therefore, destroying a specific time scale as is done with the surrogates would eliminate such interactions. This effect could be viewed as a way to investigate the presence of nonlinear dynamics related to the cross-talk among different frequencies.

Figures 3-10 are clear and explicative, but appear not very informative if taken individually, and their repetitive style may make reading the main manuscript heavier. I would suggest to move these figure in the supplementary material, and generate 2-3 figures in the main paper where the same results are presented in a more compact representation.

Eq. (8) describing the AR(2) process seems to be incorrect (p^2 should be multiplied by S_0(t-2), and an additive noise term W(t) is missing). Moreover, although it is stated that the process exhibits long-term memory, this does not appear from Eq. (8) where only short-term dependencies are modeled.

The citation of Ref. [34] is wrong/incomplete (e.g. journal name)

Page 10: a couple of sentences describing how the data were collected and pre-processed would be useful.

Minor/typos:

Page 6, “Fig. ??”

Page 9, line 5 after Eq. (9), “Fig 2, A” should be “Fig 2, panel C”

Page 12, “Bayesian model comparison ??”

Page 17, a question mark should be put at the end of the question 2 in the first paragraph of the Discussion

Page 17, “asses frequency”

Reviewer #2: This is potentially a very interesting work on the intersection of information processing, the role of cortical layers and frequency bands, with conscious states. Although I feel not qualified to review the mathematical parts of this manuscript, I believe the findings could provide new avenues for understanding (un)consciousness states induced with anaesthesia. However, in order to do so, I think the following points should be clarified or emphasized.

The authors introduce very little about the role of different cortical layers on frequencies/ information processing. This is merely stated that there are different effects across the different cortical layers, but as the main aim of the study seems to be the evaluation of the interaction between information processing, cortical layers and frequency bands, I would appreciate a more in-depth introduction of previous findings (i.e., at least including the main results of previous work). Likewise, the (minimal) differences in the findings for PFC and V1 electrodes are discussed, but the rationale for placing electrodes in these cortical areas specifically is not highlighted. It is clear that the data has been previously acquired, yet it would be good to emphasise in the introduction the logic for including/contrasting these two cortical areas based on their different (functional) cortical organization.

The material and methods section does not include sufficient information about the animal model used in the current work. A reference is provided for more information, but I encourage the authors to provide at least basic information regarding the number of recording sites per ferret (i.e., number of electrodes and number of recording sites per electrode), the isoflurane dose (and how the different doses relate to behaviour/consciousness), the determination of the point of LOC in the ferrets, the duration of the recording. Furthermore, no mention of the data pre-processing and dropout are provided. A comprehensive section should be included in the manuscript to ease interpretation of the results.

A common finding in different conditions of altered consciousness, consistent in various species, is a general slowing of the observed oscillatory activity. This has also been studied in reference 15 (in the same dataset), but no overview of these results is provided in the manuscript. It could help to place the results on AIS in context, as the reader might wonder how the results on AIS depend on potential shifts in the spectral power peaks specifically, rather than a more mechanistic role of the different cortical layers and feedforward/feedback connections. This relation between AIS and spectral power is partially addressed in Figure 11, but only for very specific delta and alpha bands, and does not account for any potential shifts in spectral power peaks. Also, do the authors have an explanation for the larger variability in normalized power at higher isoflurane doses?

In the manuscript it is stated that the mechanism of action of different anaesthetics is unknown. This is correct, the full mechanism of action of (here) isoflurane is unknown, but we do know that it acts on the GABAa receptor, glutamate receptor and alters cellular channel properties. A discussion about how these different properties could be related to alternation in information processing could help to explore the relation to (un)conscious states.

I believe the text would become more reader friendly if the literature is referred to in a more direct fashion. E.g., in the results: “However, previous work on spectral power [15, 16], as mentioned above, showed that anesthesia modulates cortical areas and layers differently.”, then it would be nice to read how exactly it differed. In the discussion: “Even though the molecular targets of anesthetic agents are well known [42],”, then I suggest the authors to at least mention the targets of isoflurane.

Minor comments:

-Please mention in the abstract that this is a study in ferrets, not in human.

-The axes of Figure 11D refer to delta, the text refers to alpha.

-Author summary missed a word: Here -we- introduce such a measure and study how isoflurane anesthesia

affects the local information processing in the ferret prefrontal and primary visual areas

around loss of consciousness.

-Discussion misses a citation at “only a handful of studies exist to date [?,6-8, 13],”

-Discussion misses a word: where activity -in- supragranular layers

-Hz is sometimes written italic and sometimes not.

Reviewer #3: In this paper, Pinzuti et al. tackle the problem of forming a spectrally resolved estimator of active information storage (AIS), and using it to characterise some aspects of neural computation in data from ferrets undergoing general anaesthesia. The authors are able to formulate a spectral AIS estimator thanks to a direct application of a previous method, and apply it to a rich dataset of LFP recordings across cortical regions and layers. The explicit focus on linking information processing with features of neural activity (oscillations) is interesting; and I commend the authors for running their statistical analysis with the powerful pymc3 library as well as for their use of Bayes factors for analysis.

Overall, I find the subject of the paper very interesting and worth pursuing. However, as it stands, the method isn't particularly novel by itself, and a clearer presentation and interpretation of the results is sorely lacking. For these reasons, and despite the great potential of this line of work, I cannot recommend the paper for publication without substantial revisions.

## Major comments (most important first)

1) Results and interpretation

The results of the AIS analysis of ferret LFP data are spread across a lengthy description and 8 figures. The results are described in great detail, but without any clear summary or clear interpretation of what they mean for the neuroscience of anaesthesia (or the neuroscience of consciousness more generally). With a few minor exceptions, the authors rarely draw specific conclusions showing how the spectral AIS yields new insights about cortical dynamics under anaesthesia, and rely on rather vague statements (e.g. alluding to the "different computational properties of the layers" [p.3], but without specifying what these properties might be).

This makes it really hard for the reader to build a coherent picture of the results, other than "results varied widely between bands" and "lower frequencies are more spatially homogeneous." To be clear: I don't mean to say that the results are meaningless or can't be interpreted; I mean that with the current presentation it's extremely difficult for the reader to make sense of so much information.

I would strongly encourage the authors to make a smaller number of more representative figures that can convey the results in a more convincing way. While I am a fan of detailed statistical reporting, I see no reason why all ~90 posteriors should be in the main text instead of in the supplementary material. A clearly presented analysis of spectral AIS and its relationship to anaesthesia/consciousness in the different layers and regions, accompanied by a strong theoretical argument or perhaps even a computational model, would greatly strengthen the paper.

2) Meaning of the test

Overall, the statistical properties of the spectral AIS are a bit unclear. The examples are very useful in showcasing spectral AIS in systems with clear narrow-band activity, but this leaves many questions unanswered:

- What is precisely the null hypothesis behind the null distribution? Can it be formalised mathematically?

- The authors demonstrate the test is sensitive to information storage in a particular band, by showing that it successfully identifies processes with activity in the band of interest. However, they do not show that the test is specific, which they should do by showing the test finds no significant effects in a system with AIS in all bands except the band of interest. (Or, even better, in a system with high spectral power but no AIS in the target frequency band, if this is possible, which would very strongly emphasise the authors' point that spectral AIS is meaningfully different from power.)

- In Eq. 7, why is the future state taken from the unshuffled process?

- Perhaps most importantly, what is the meaning of surrogates having higher AIS than the original signals? These cases are clearly very significant (e.g. in Fig. 1), yet they are almost completely left unmentioned in the text, with the exception of some unproven statements that "only positive values correspond to a significant contribution to the formation of AIS."

- In multiple places (e.g. page 18) the authors refer to the proposed method as a frequency decomposition. But, in what sense is it a decomposition? It doesn't seem to me that AIS_freq meaningfully decomposes the time-domain AIS (compare to e.g. PID, which can be called a decomposition because all atoms sum up to the quantity that is being decomposed, namely the joint mutual information).

Finally, the comparisons between spectral power and spectral AIS were very interesting, and a crucial part of the authors' argument that spectral AIS reveals new information that was not available with previous tools. Unfortunately, this subsection is the shortest one in the Results section, and these relationships between spectral AIS and other methods were not investigated for the three example systems.

Overall, I would encourage the authors to clarify the statistical properties of their estimator (especially the issue of surrogates having higher AIS), and explore more systematically the relationship with spectral power.

3) Novelty

The method used in this paper is a direct application of a previously published method by the authors in this same journal. Substantial fractions of the text and Figure 1 are directly adapted from this previous paper. The data has also been analysed multiple times in previous publications, including a paper by some of the authors using AIS, as well as another one using the authors' spectrally resolved transfer entropy (TE) measure. Therefore, the potential for novel contributions of the paper is rather limited.

While I do not see this as a critical flaw for publication, it does mean that the authors should do significant additional work to satisfy major comments 1 and 2 above and clearly show the novel contributions of this manuscript.

## Minor comments

- The paper has several minor mistakes and typos throughout that should be addressed. As a non-exhaustive list of examples: p. 16, "frequencies range" (should be "frequency ranges"); p. 18, "Hz" is inconsistently written both with and without italics; p. 14, undefined reference; p. 16 "as the power increase" (should be "as the power increases"); etc).

- I find the notation is at times inconsistent. For example, N is used to denote both the length of the time series and the number of surrogates. In page 7, it seems W_{J_0} represents all wavelet coefficients, while from the definition in page 5 it would seem it is only the coefficients for the J_0 scale. Vector notation is also inconsistent, as Y is written both with and without bold font in page 4, and X is written without bold font. Specifically on the topic of vector notation, I think it would be much clearer for the reader if the authors were to specify the size of all matrices and vectors involved as they are introduced.

- In page 6, the authors state that "with this method, we are also able to handle any potential bias introduced by the wavelet filtering of the surrogate data." What does this mean? Could you elaborate?

- Page 7: Should AIS' be indexed by j? It depends on the wavelet scale, if I understand correctly.

- Paragraph after Eq. 9: should it be Fig 2C, instead of 2A?

- Example III: the paragraph after Eq. 9 says that w_1,w_2 "are the natural frequencies of the oscillators which were set to 0.8 and 0.9." However, later it says that the process "oscillated around 8 Hz." How are these related? And what's the sampling rate of the process?

- Page 10: Remove the "0.0.1" numbering in the section header (same with "0.0.2" in page 16).

- Page 11: instead of "model squared," it would be more consistent with statistical literature to call it "quadratic model."

- Page 11: I suspect there's something missing after "we report the full modeling and model comparison results in..."

- Page 16 and Fig. 11: I suspect something may have been misplaced in the figure, because the BFs reported in the text do not match those in the figure. For example, the first reported BF is "BF= 48530.5 (see Fig 11, panel A, top row)," while that panel has BF = 1.380.

- How are the trials from both ferrets pooled together for analysis?

**Have the authors made all data and (if applicable) computational code underlying the findings in their manuscript fully available?**

Reviewer #1: **No: **

Reviewer #2: **No: **Data is not available, the code is

Reviewer #3: Yes

PLOS authors have the option to publish the peer review history of their article (what does this mean?). If published, this will include your full peer review and any attached files.

Reviewer #1: **Yes: **Luca Faes

Reviewer #2: No

Reviewer #3: No
---

## [Decision Letter · Decision Letter 1]

5 Jan 2023

Dear Mr Pinzuti,

We are pleased to inform you that your manuscript 'Information theoretic evidence for layer- and frequency-specific changes in cortical information processing under anesthesia' has been provisionally accepted for publication in PLOS Computational Biology.

Before your manuscript can be formally accepted you will need to complete some formatting changes, which you will receive in a follow up email. At that stage you can also update one reference as noted by one of the reviewers. A member of our team will be in touch with a set of requests.

Best regards,

Daniele Marinazzo

Section Editor

PLOS Computational Biology

Reviewer's Responses to Questions

**Comments to the Authors:**

Reviewer #1: All my comments are adequately addressed in the revised paper. I recommend acceptance of the manuscript as is.

Minor: Ref. [75] should be substituted by the journal publication: ANTONACCI, Yuri, et al. Measuring High-Order Interactions in Rhythmic Processes through Multivariate Spectral Information Decomposition. IEEE Access, 2021, 9: 149486-149505.

Reviewer #3: The authors have addressed the reviewers' concerns to an acceptable degree. The paper is in good enough condition for publication.

**Have the authors made all data and (if applicable) computational code underlying the findings in their manuscript fully available?**

Reviewer #1: Yes

Reviewer #3: Yes

PLOS authors have the option to publish the peer review history of their article (what does this mean?). If published, this will include your full peer review and any attached files.

Reviewer #1: No

Reviewer #3: No

---

## [Editor Report · Acceptance letter]

23 Jan 2023

PCOMPBIOL-D-22-01077R1 

Information theoretic evidence for layer- and frequency-specific changes in cortical information processing under anesthesia

Dear Dr Pinzuti,

I am pleased to inform you that your manuscript has been formally accepted for publication in PLOS Computational Biology. Your manuscript is now with our production department and you will be notified of the publication date in due course.

With kind regards,

Zsofia Freund
